# Experimental demonstration of third-order memristor-based artificial sensory nervous system for neuro-inspired robotics

See-On Park ®[1], Hakcheon Jeong ®[1], Seokho Seo ®[1], Youna Kwon[2], Jongwon Lee ®[3] ✉ & Shinhyun Choi ®[1,4] ✉

The sensory nervous system in animals enables the perception of external stimuli. Developing an artificial sensory nervous system has been widely conducted to realize neuro-inspired robots capable of effectively responding to external stimuli. However, it remains challenging to develop artificial sensory nervous systems that possess sophisticated biological functions, such as habituation and sensitization, enabling efficient responses without bulky peripheral circuitry. Here, we introduce a memristor device with third-order switching complexity, emulating an artificial synapse that inherently possesses habituation and sensitization properties. Incorporating an additional resistive switching $TiO_x$ layer into the $HfO_2$ memristor exhibits third-order switching complexity and non-volatile habituation characteristics. Based on the third-order memristor, we propose a robotic system equipped with a memristor-based artificial sensory nervous system for optimizing the robot arm's response to external stimuli without the aid of processors. It is experimentally demonstrated that the robot arm with the developed memristor-based artificial sensory nervous system ignores approximately 71% of safe and familiar stimuli while sensitively responding to threatening and significant stimuli, similar to the habituation and sensitization of biological sensory nervous systems. Our findings can be a stepping stone for energy-efficient and intelligent robotic systems with reduced hardware burden.

Animals, including cats, birds, and humans, instinctively adapt their responses to external stimuli based on their surrounding environment or previously learned information. For instance, a dog accustomed to wearing clothes may not react to the tactile stimulus of the fabric, having learned that it is not harmful. Similarly, a newborn baby might initially startle at the sound of a refrigerator, but the baby becomes desensitized after learning that the noise is not a threat. These adjustments in response to stimuli occur unconsciously, facilitated by the sensory nervous system (SNS) in animals[1,2].

The SNS receives external stimuli, converts them into electrical signals, and transmits these signals to the central nervous system. This process enables animals to perceive their surroundings. The SNS is essential for survival, as it not only detects external stimuli but also promptly processes information, facilitating rapid responses without direct involvement from the central nervous system. For example, the SNS modulates sensitivity to specific stimuli, influencing the intensity of an animal's reaction to them. One of these synaptic modulations, called habituation, is achieved by reducing the synaptic strength of

[1]School of Electrical Engineering, Korea Advanced Institute of Science and Technology (KAIST), Daejeon, Republic of Korea. [2]Nano Convergence Technology Division, National Nanofab Center (NNFC), Daejeon, Republic of Korea. [3]Department of Semiconductor Convergence, Chungnam National University, Daejeon, Republic of Korea. [4]Graduate School of Semiconductor Technology, Korea Advanced Institute of Science and Technology (KAIST), Daejeon, Republic of Korea. ✉e-mail: jwlee80@cnu.ac.kr; shinhyun@kaist.ac.kr

involved neurons, resulting in a decreased response. Habituation enables animals to disregard familiar and non-threatening signals[3-8]. On the contrary, an increase in synaptic strength, known as sensitization (or dishabituation if it reverses the effects of prior habituation)[5], amplifies responses. This makes animals more responsive to potentially threatening and significant stimuli. Therefore, habituation and sensitization serve as crucial functions for survival, conserving energy by filtering out irrelevant responses to harmless stimuli and enabling animals to concentrate on responding to significant signals.

Unlike animals with biological SNS, robotic systems receiving various external stimuli from the environment are overwhelmed by the vast data generated by sensors, given the limited performance of their processors. Implementing biological SNS functions, such as habituation and sensitization to filter out insignificant stimuli at the edge levels, has been considered an effective solution to reduce the burden on processors from the vast amount of data. Inspired by the efficiency of the biological SNS, there has been a surge of interest in developing robotic systems that mimic these functions. To implement the habituation and sensitization in robots, an artificial sensory nervous system (ASNS) had been developed based on complementary-metal-oxide-semiconductor (CMOS) circuitries[9-11]. The CMOS-based ASNS accurately emulates the habituation and sensitization of the biological SNS, however, it requires a large area for implementing the CMOS circuitry and consumes much higher energy compared to the biological SNS, which limits the realization of robotic systems with habituation and sensitization.

Instead of the CMOS-based ASNS, memristors have attracted attention as a breakthrough for neuro-inspired robotics[12-22]. A memristor, an emerging two-terminal non-volatile memory device, stores analog information by forming or rupturing an internal conductive filament, resulting in changes in device resistance between a high resistance state (HRS) and low resistance state (LRS)[23-26]. Based on the analog memory property, fast speed, high scalability, and structural simplicity, the memristor has been utilized as an energy-efficient artificial synapse device in various fields such as neuromorphic computing or neuro-inspired robotics.

Utilizing the analog synaptic properties of the memristor, several memristor-based artificial sensory nervous systems (MASNSs) have replicated key functions of the biological SNS. These functions include long-term potentiation and depression (LTP and LTD, respectively), spike-time-dependent plasticity (STDP), synaptic adaptation, and associative learning[12,15,18]. Despite these advancements, the practical implementation of a robotic MASNS exhibiting both habituation and sensitization capabilities has not yet been experimentally demonstrated. This gap is attributed to the resistive switching in memristors, which is typically governed by one or two state variables, resulting in a simplistic switching behavior insufficient for replicating more complex functions such as habituation and sensitization[27]. Therefore, developing a memristor that not only offers non-volatile SNS functionalities—such as habituation and sensitization—but also meets high performance standards is crucial for the advancement of efficient robotic systems equipped with MASNS.

In this study, we emulated the complex behavior of a biological synapse such as habituation, sensitization, and frequency-dependent plasticity with the developed third-order memristor having three state variables (filament diameter, device temperature, and $TiO_x$ conductivity). The developed memristor also exhibited high performance of $HfO_2$-based memristors such as non-volatile memory, good analog property, and low variations. By utilizing the third-order memristor, we developed a robot arm system with a MASNS. This system is capable of sensing tactile and electric shock stimuli. It suppresses responses to non-threatening stimulus, while specifically reacting to dangerous electric shock stimuli. Based on the habituation and sensitization characteristics of the third-order memristor, the MASNS-implemented robot arm showed efficient response to external stimuli by ignoring

approximately 71% of safe stimuli while responding to every identified threatening electric shock (harmful) stimulus. In contrast, conventional low-order (first- or second-order) memristor-based MASNS system have ignored only 0.2% of safe stimuli. Our results open up the possibility of an intelligent neuro-inspired robotic system that optimizes its responses to the environment without bulky processors, and therefore, can be utilized for energy-efficient robots operating in a power-limited environment or rescue robots which ignore unimportant stimuli for fast operations.

## Results

### Habituation and sensitization of a biological sensory nervous system

Habituation and sensitization are vital functions for the survival of animals, as they enable animals to focus only on significant stimuli and prevent the waste of energy on unnecessary responses to insignificant stimuli. Examples of habituation and sensitization are illustrated in Fig. 1a. For instance, when a mouse hears an unfamiliar twitter from a small bird, it will sensitively respond (e.g., by running away or hiding) because it does not know if the sound comes from predators. However, if the same twittering from the small bird is repeated without any threat, the mouse recognizes the twittering as safe and no longer pays attention to it. This process is called habituation, and it is the result of synaptic modulations in the SNS of the mouse; for example, the synaptic strengths of some synapses related to the twittering are reduced as the safe twittering is repeated, and the neurons will be less activated. Consequently, by ignoring the twittering, the mouse saves time and energy that would otherwise be wasted for running away.

On the contrary, if the mouse, having already undergone habituation, encounters an owl twittering similar to that of the little bird, it initially ignores the twittering due to the habituation. However, after the owl repeatedly attacks the mouse, the mouse sensitively responds to the twittering, even if it comes from the safe little bird. The enhanced sensitivity to specific stimuli allows the mouse to sensitively respond to threats and survive from predators. This process is called sensitization (or dishabituation when habituation precedes it), resulting from the enhancement of related synaptic connections. The response intensity and synaptic strength regarding specific stimuli can be illustrated as shown in Fig. 1b. The habituation occurs when a stimulus (e.g., tactile) is repeatedly received without threats (e.g., pain). Sensitization, on the contrary, occurs when the stimulus is accompanied by threats, and both habituation and sensitization are the results of synaptic strength modulation in the SNS.

To realize an ASNS in a robotic system, memristors have been widely utilized as artificial synapses of the ASNS in robots. A diagram of the MASNS is illustrated in Fig. 1c. A pre-synaptic neuron senses stimuli and transfers signals to the memristor, the memristor passes different outputs depending on its conductivity (synaptic strength), and the post-synaptic neuron makes locomotion of the robot if the signal from the memristor exceeds a threshold[20]. Realistically mimicking the functions of the biological SNS requires memristors capable of replicating the complex characteristics of biological synapses.

For habituation function, the conductance needs to initially increase, and then decrease after a few initial set pulses (blue line of Fig. 1d). However, only one or two state variables, a filament diameter and device temperature, have been demonstrated for several non-volatile memristors including oxide-based memristors and metal cation-based memristors[27-31]. These low-order memristors have been hardly used for implementing the habituation of the biological SNS, because the device conductance usually monotonically increases with set pulses (red line of Fig.1d and Supplementary Fig. S1). Furthermore, the habituation state needs to be stably maintained (non-volatile) for a long time, even after the applied pulses are withheld for a while (see Supplementary Fig. S2). Therefore, a new type of memristor with third-order dynamics, implying a device with another state variable beyond

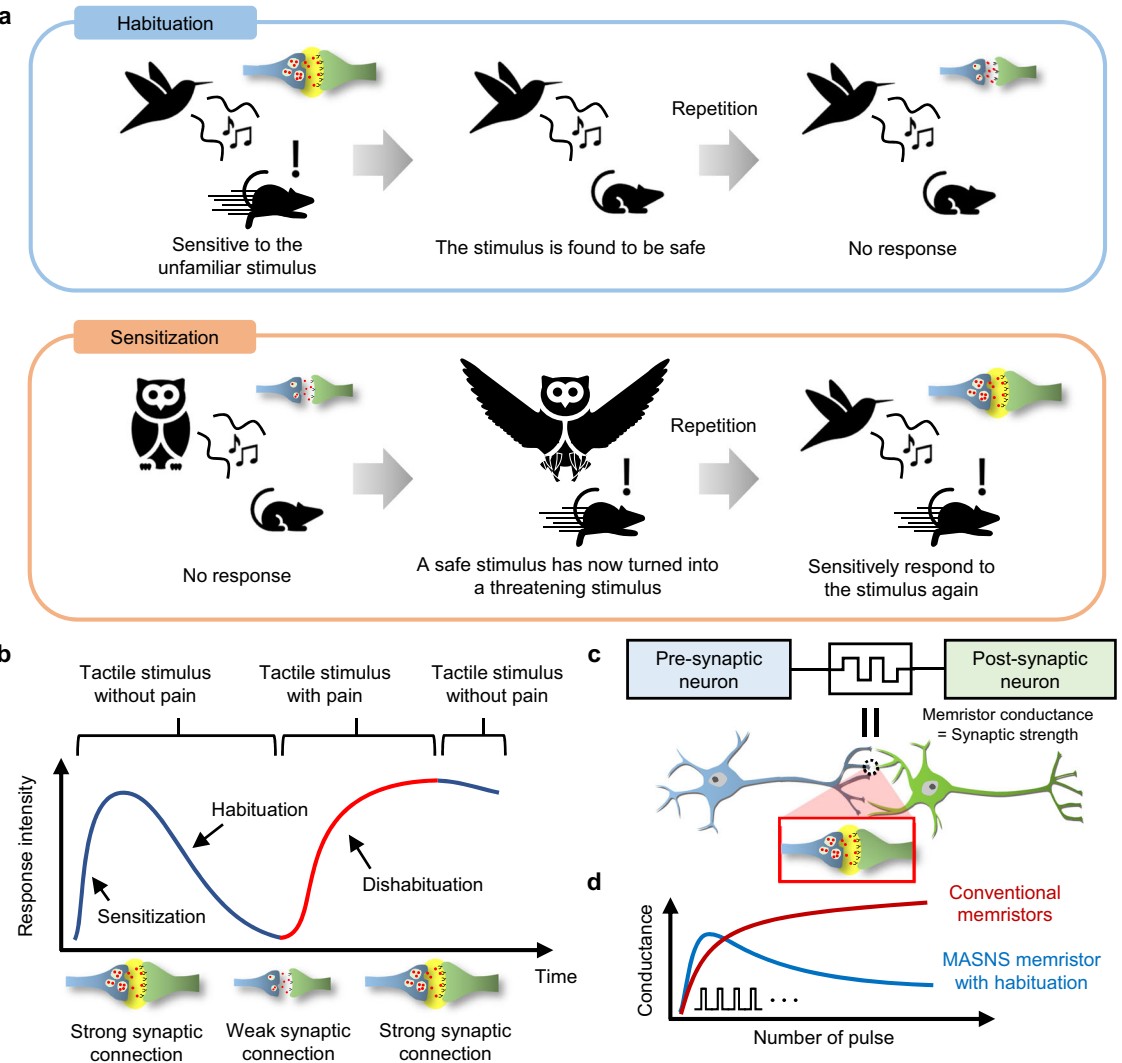

**Fig. 1 | Habituation and sensitization of the SNS. a** Illustrations of habituation and sensitization. **b** The change of the response intensity and the synaptic strength according to the applied stimulus. The enhancement of the synaptic connection induces sensitization while the reduction of the synaptic strength induces habituation. **c** The MASNS and the biological SNS. In the MASNS, the conductance of the memristor determines the synaptic strength between the pre-synaptic and post-synaptic neurons. **d** The conductance update curve of memristors for MASNS (blue) compared to conventional low-order memristors (red).

the filament diameter and the device temperature, is required to imitate those complex behaviors of biological synapses (see Supplementary Fig. S3).

### Third-order and CMOS-compatible HfO₂ memristor

Similar to other filamentary memristors, $HfO_2$ memristors modulate the device conductance by controlling the internal conductive oxygen vacancy-based filament[32–34]. Various $HfO_2$ memristors have demonstrated favorable characteristics for analog artificial synapses, including high endurance, fast speed, high yield, and good uniformity[25,32,35–37]. However, only first- or second-order switching (state variable: filament diameter and the device temperature) has been demonstrated for $HfO_2$ memristors[28], whereas third-order switching has not been explored yet.

To develop a higher-order $HfO_2$-based memristor with good switching performances, an additional conductive $TiO_x$ layer is inserted in the $HfO_2$ memristor. Because of the high reactivity of the titanium to the oxygen, we confirmed that the resistive switching $TiO_x$ layer is formed between the Ti and $HfO_2$ layers during the back-end-of-line (BEOL) fabrication process. The fully CMOS-compatible third-

order memristor having $TiN/Ti/TiO_x/HfO_2/TiN$ stack is fabricated on an 8-inch wafer as shown in Fig. 2a–c and Supplementary Fig. S4. More detailed information about the device fabrication is in the Methods section. The fabricated $HfO_2$ memristor exhibited favorable operation characteristics with stable retention, fast switching speed, low switching variations, and good endurance (Supplementary Fig. S5, S6, and Supplementary Table S1). Furthermore, the high yield within 8-inch large-scale CMOS-compatible fabrication (Supplementary Figs. S7 and S8) and high thermal stability in elevated temperatures (Supplementary Fig. S9) enable integration of the device with sensor arrays and make the device suitable for reliable robotic applications in various environments.

To understand the third-order resistive switching behavior based on the device structure, transmission electron microscopy (TEM) imaging, electron dispersion spectroscopy (EDS), X-ray photoelectron spectroscopy (XPS) results were obtained, as shown in Figs. 2b, c, and Supplementary Fig. S10. The TEM image in Fig. 2b shows metal-insulator-metal structure of the fabricated device, where the device consists of CMOS-compatible materials including the TiN, $HfO_2$, or Ti. It is noted that the device is fabricated in fully CMOS-compatible

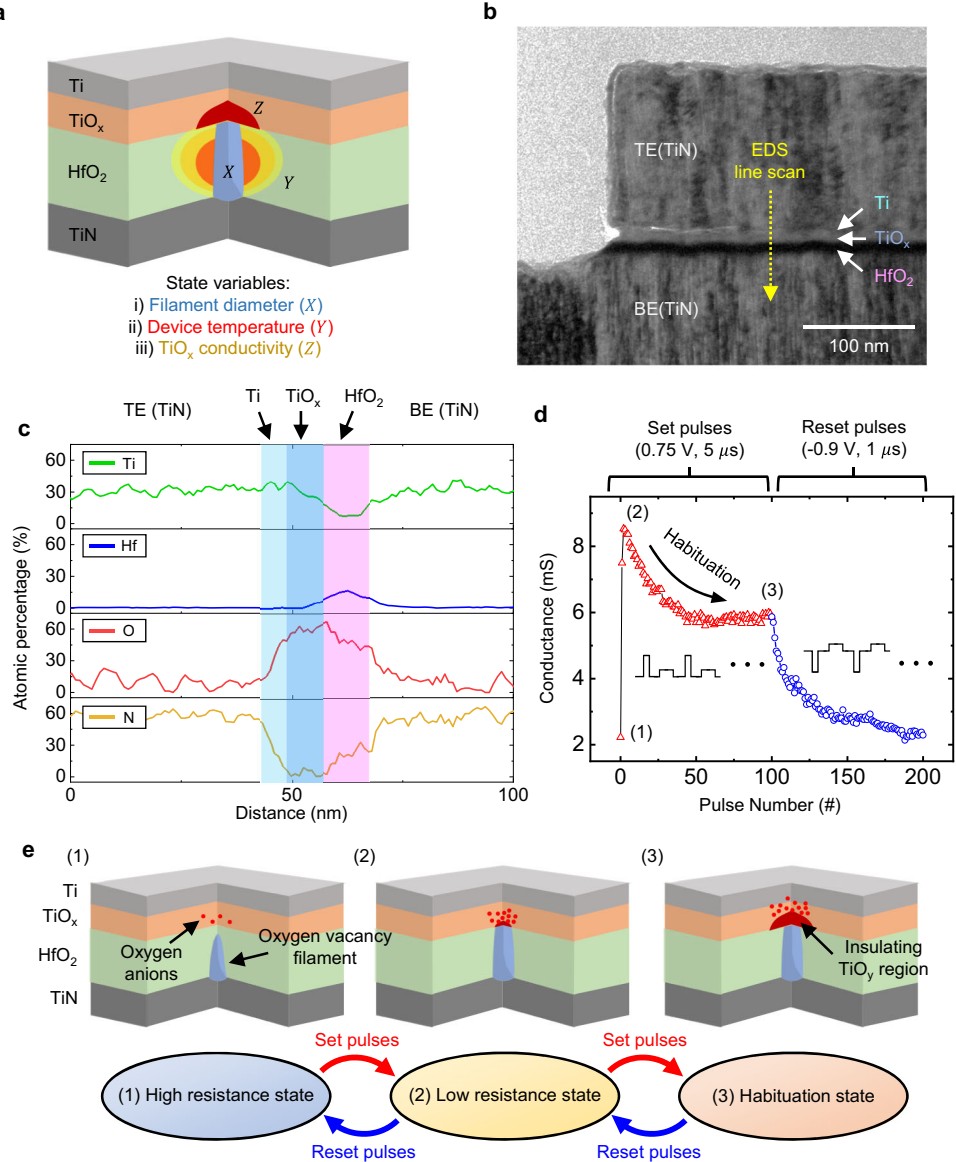

**Fig. 2 | The developed third-order memristor for habituation and sensitization. a** The schematic of the fabricated third-order memristor and the state variables. Three state variables: filament diameter, device temperature, and $TiO_x$ conductivity. **b** TEM image of the third-order memristor. **c** The EDS line-scan results of the third-order memristor confirming that the $TiO_x$ layer between the Ti and $HfO_2$ layer for another state variable. **d** The conductance update curve of the third-order memristor during consecutive 100 positive voltage pulses followed by 100 negative voltage pulses. After the sensitization has occurred for the first two positive set voltage pulses, the habituation (decrease of the conductance) is observed during the set pulses. The negative reset voltage pulses initialize the device and reduce the device conductance into the HRS. The conductance state is read by a read pulse (0.3 V and 50 μs) after each set and reset pulse. **e** Illustrations of the third-order memristor during the voltage pulses and the corresponding conductance states.

environments, which enables the monolithic integration of the device to various CMOS peripheral circuitries. The EDS line scan results, as shown in Fig. 2c, show the Ti/$TiO_x$/$HfO_2$ stack with an oxidized Ti-$HfO_2$ interface between the TiN electrodes. The initial stoichiometry of the $TiO_x$ layer is close to TiO, as demonstrated by the XPS results (see Supplementary Fig. S10).

The fabricated device possesses at least three state variables, including not only the filament diameter and device temperature in $HfO_2$-based memristor but also the $TiO_x$ layer conductivity. In this device, the $TiO_x$ layer provides an additional state variable (resistive switching in the $TiO_x$ layer) compared to conventional oxide-based memristors, enabling third-order memristor behaviors. The filament diameter and $TiO_x$ conductance inversely affect the device's total conductance in response to set pulses. The device without the $TiO_x$

layer exhibited typical monotonic conductance update curve of conventional low-order memristors, demonstrating that the $TiO_x$ layer is essential for habituation characteristics (see Supplementary Fig. S11).

When the device is in the HRS, the device conductance initially increases to the LRS (the state (1)–(2) of Fig. 2d, e) as set pulses are applied, since the redistributed oxygen anions from the $HfO_2$ layer to the $TiO_x$ layer lead to an increase in the filament diameter. However, as more set pulses are applied, more oxygen anions are drifted to the $TiO_x$ layer. The redistributed oxygen anions change the initially conductive $TiO_x$ layer near the filament to an insulating $TiO_y$ region (the state (2)–(3) of Fig. 2d, e). The insulating $TiO_y$ region has higher resistance compared to the conductive $TiO_x$ layer, and therefore, the device conductance decreases to habituation state as set pulses are applied. The device conductance state can be programmed back into

the HRS by applying strong negative bias reset pulses or back into the LRS by applying weak negative bias reset pulses (see Fig. 2d and Supplementary Fig. S12).

Conductance update curves of the device under various pulse conditions demonstrated that the third-order switching behavior requires a high amplitude or long width voltage pulse to relocate oxygen anions into the TiO$_x$ layer (Supplementary Fig. S13 and Supplementary Table S2). Furthermore, we investigated whether the habituation characteristics originate from the device's resistive switching or the transient and parasitic effects, such as intrinsic capacitance or current overshoot. By analyzing the intrinsic capacitance of the device and varying the pulse interval to mitigate current overshoot, as shown in Supplementary Figs. S14 and S15, it was confirmed that the habituation characteristics arise from the resistive switching in the device, rather than from transient or parasitic effects.

There have been several attempts to emulate habituation characteristics using memristors[16,38–41]. For instance, memristors employing a unipolar reset process[16] (such as filament rupture due to Joule heating) or circuits composed of volatile memory devices[38–41] have exhibited habituation characteristics in response to repeated stimuli. However, these systems exhibited volatile habituation characteristics, indicating that the trained information was hardly maintained over time. In contrast to the volatile habituation characteristics, each state (state (1), (2), and (3) of Fig. 2d, e) of the third-order HfO$_2$ memristor is not a transient conductance fluctuation, but is a non-volatile memory state, as shown in Supplementary Fig. S16. The stable and non-volatile habituation can be realized by implementing the third-state variable (TiO$_x$ conductivity), which provides a distinct non-volatile memory state (habituation) other than the HRS and LRS (see Fig. 2f).

The effect of the device temperature as a second state variable was also investigated in Supplementary Fig. S17 and S18. To demonstrate the impact of device temperature on conductance update, a consecutive 100 voltage pulse train with varying pulse intervals was applied to the third-order memristor. As shown in Supplementary Fig. S17, the habituation characteristic became evident as the pulse interval shortens, where a shorter pulse interval results in less heat dissipation between pulses, leading to an increase in device temperature. The elevated heat energy enhances the drift of the oxygen anions into the TiO$_x$ layer, forming the insulative TiO$_y$. In addition, measurements at elevated temperatures revealed more pronounced and fast habituation characteristics, demonstrating the effects of the high temperature on resistive switching in the TiO$_x$ layer and habituation characteristics (see Supplementary Fig. S18). The results support the conclusion that the device temperature acts as a critical state variable in the developed third-order memristor.

It is noteworthy that there are some discrepancies between the habituation of the third-order memristor and that of biological synapses. While biological synapses exhibit spontaneous recovery due to short-term memory, the third-order memristor features non-volatile memory. The use of non-volatile memory involves a trade-off between synaptic weight stability and biological plausibility. The absence of short-term memory may limit the use of the third-order memristor for accurately emulating synaptic behaviors for synaptic learnings. However, reliable robotic applications often require learned information to persist for a long time or even when powered off, making non-volatile memory advantageous. Despite the absence of short-term memory, the third-order memristor demonstrated various synaptic behaviors associated with habituation, as shown in Supplementary Figs. S19, S20, and Supplementary Note S1. Furthermore, certain biological synaptic behaviors reliant on short-term memory, such as spontaneous recovery and potentiation of habituation, can be mimicked by periodically applying weak reset pulses, if necessary (see Supplementary Figs. S19g, S19h, and S20). Detailed comparisons of the third-order memristor with several devices emulating habituation, as presented in Supplementary Table S1, highlight its favorable electrical characteristics for

reliable robotic applications, as well as ability to mimic various synaptic behaviors.

## Operation principle of the third-order memristor

The switching mechanism in the third-order memristor is further supported by modeling the conductance update curves of the filament and the TiO$_x$ layer, respectively (see "Methods"). As shown in the Fig. 3a, the third-order memristor is modelled with two anti-serially connected resistive switching variables (representing the filament and TiO$_x$ layer, respectively). Based on the model, we obtained the total simulated output current and compared it to the experimental data (Fig. 3b–m). The typical LTP and LTD are obtained for the shorter set pulse widths (1 and 2 μs), while the habituation characteristic is observed for the longer set pulse widths (3 and 4 μs). It matches well with the experimental data of the third-order memristor. The identical non-monotonic conductance update curve from the model, which is hardly acquired in conventional low-order memristors, demonstrates that the habituation characteristics originate from the anti-serially connected resistive switching filament and TiO$_x$ layer.

To further disclose the switching mechanism of the third-order memristor, the conduction mechanisms for each resistance state (HRS, intermediate state between HRS and LRS, LRS, and habituation state) were investigated, as shown in Supplementary Fig. S21. When the device was in HRS, intermediate state, or LRS, the linear $I–V$ relationship of hopping conduction was confirmed (see Supplementary Fig. S21b). On the contrary, for the habituation state, the $I–V$ curve exhibited a non-linear relationship, indicating that the Schottky emission dominates conduction in the habituation state (see Supplementary Fig. S21c). The different conduction mechanisms for each state can be explained by the initially conductive and defective TiO$_x$ layer, where the oxygen anions from the filament reversibly oxidize the defective TiO$_x$ layer into the resistive TiO$_y$ layer with fewer defects, resulting in the Schottky emission (see Supplementary Figs. S21d–g). The results demonstrate that the movement of oxygen anions and the reversible oxidation of the TiO$_x$ layer induce the habituation characteristic.

The effect of the HfO$_2$ layer thickness was investigated, as shown in Supplementary Fig. S22. The devices with thin HfO$_2$ layer (5 nm) and thick HfO$_2$ layer (10 nm) were fabricated and their resistive switching characteristics were compared by applying consecutive 100 set pulses followed by 100 reset pulses. As shown in Supplementary Fig. S22a, the device with a thin HfO$_2$ layer did not show habituation regardless of the set pulse amplitude. The on/off ratio of the resistive switching increases as the set voltage pulse amplitude increases, and the device breaks down when the set voltage pulse amplitude exceeds 0.8 V. In the thinner HfO$_2$ device, the oxygen anions from the thin HfO$_2$ layer are not enough to form an insulative TiO$_y$ region in the TiO$_x$ layer. The thin HfO$_2$ device showed typical LTP and LTD of the low-order memristor while the thick HfO$_2$ device exhibited the habituation characteristic (see Supplementary Figs. S22b, c). Therefore, the third-state variable, the TiO$_x$ layer conductance, is not available for the thinner HfO$_2$ device, indicating that the thickness of the HfO$_2$ layer affects the resistive switching in the TiO$_x$ layer.

## Memristor-based artificial sensory nervous system with habituation and sensitization

In the biological SNS, a post-synaptic motor neuron, responsible for determining animal responses, is connected to pre-synaptic sensory neurons through synapses. The pre-synaptic sensory neurons perceive different types of stimuli such as touch and pain. They transfer output spikes to the post-synaptic motor neuron when stimuli are received (see Fig. 4a).

Once the safe touch stimulus is received repeatedly without the pain stimulus, the synaptic strength between pre-synaptic and post-synaptic neurons decreases. It leads to gradually reduced activity of

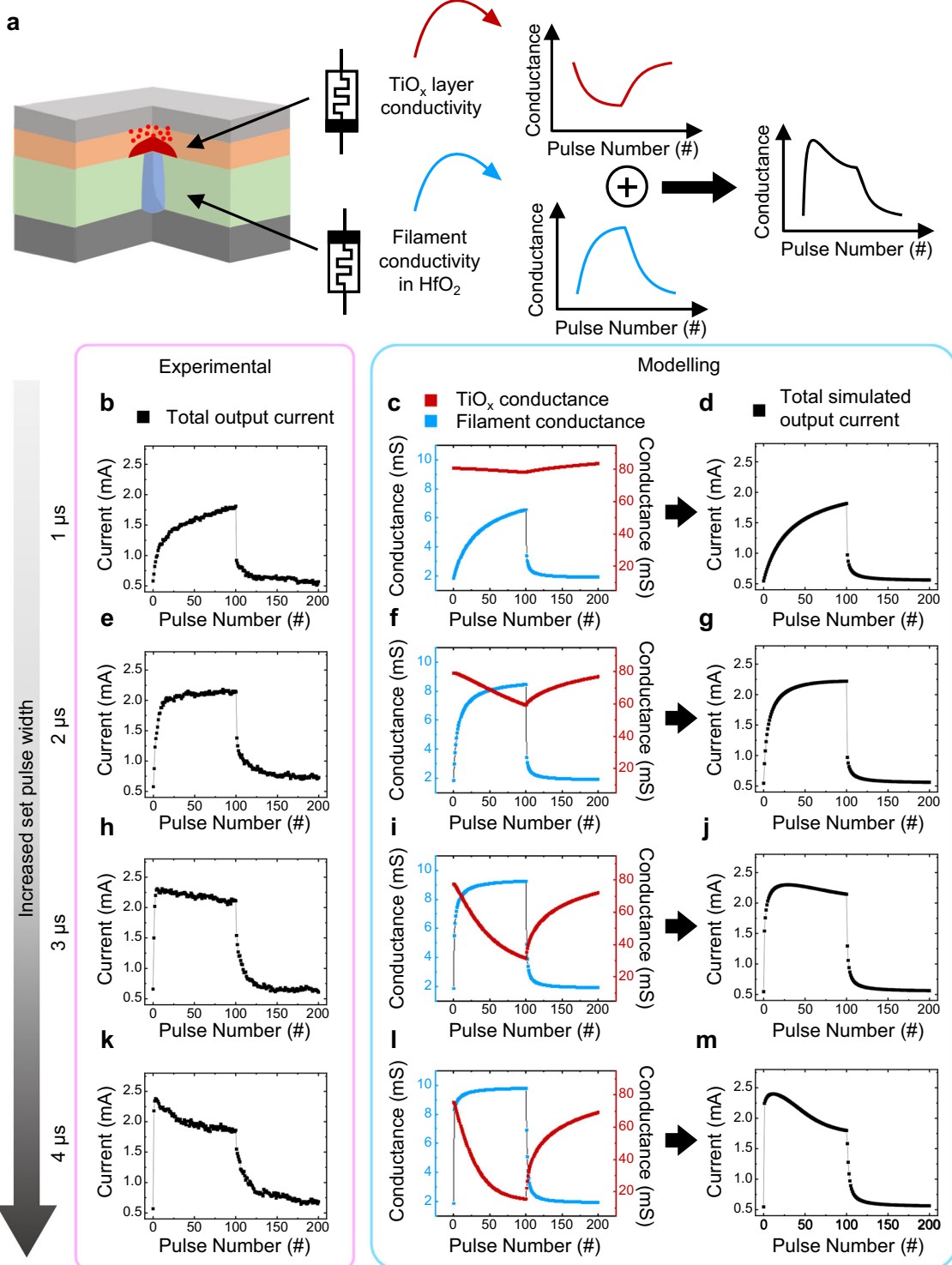

**Fig. 3 | Resistive switching model with the anti-serially connected TiO$_x$ layer and filament. a** Illustrations of the conductance updates of the TiO$_x$ layer and filament in the third-order memristor. The TiO$_x$ layer and filament show opposite conductance update curves to the applied voltage pulse due to the anti-serially connected structure, which enables habituation characteristics. **b**–**m** Comparisons between the experimental data and the simulated results. By modelling the conductance update curves of the TiO$_x$ layer (red) and the filament (sky blue) respectively (**c**, **f**, **i**, **l**), the total simulated output currents of the third-order

memristor with various set pulse widths are obtained (**d**, **g**, **j**, **m**). The total simulated output current exhibit the similar conductance update curves to the experimental data (**b**, **e**, **h**, **k**), demonstrating that the opposite resistive switching of the filament and TiO$_x$ layer induce habituation characteristic of the third-order memristor. The experimental conductance update curves in (**b**, **e**, **h**, **k**) were measured by consecutive 100 set pulses (0.75 V) with various widths (1 to 4 μs) followed by 100 reset pulses (−0.9 V and 1 μs).

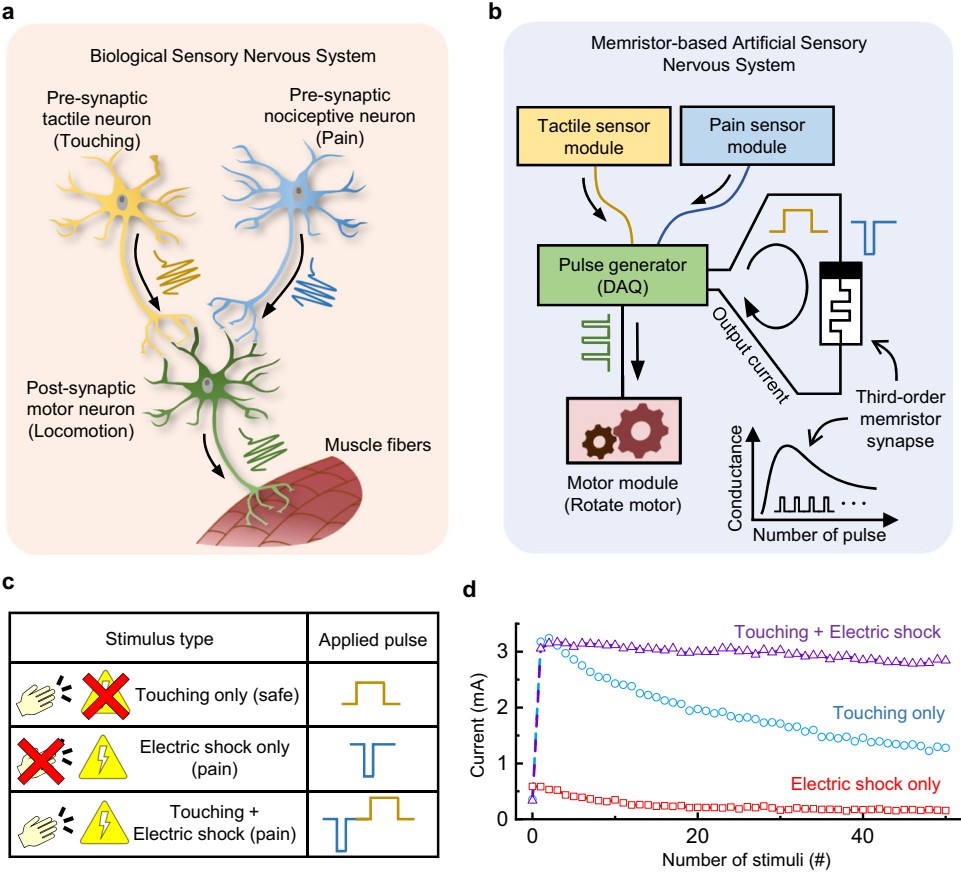

**Fig. 4 | Schematic of the MASNS and its habituation behavior to the safe stimulus. a** Schematic of the biological SNS. The pre-synaptic tactile neuron sensing a touch stimulus (yellow) and the pre-synaptic nociceptor sensing a pain stimulus (blue) are connected to the post-synaptic motor neuron (green), which generates responses. **b** Schematic of the developed MASNS for habituation and sensitization. **c** The voltage pulse forms according to the stimulus types. **d** The conductance update curves of the third-order memristor in the MASNS when various stimuli (touching only (0.8 V and 1 μs), electric shock only (−0.9 V and 1 μs), and both of the touching and the electric shock) are applied 50 times. The MASNS gradually reduces its sensitivity when the touching only stimuli are applied, while it maintains high sensitivity when the touching and the pain are applied together. The MASNS maintains low sensitivity for the electric shock only case because the pain is not related to the other stimulus. The device state was read by a read pulse (0.3 V and 100 μs).

the post-synaptic motor neuron due to the habituation. On the contrary, when touch and pain stimuli are received simultaneously, the synaptic strength between pre-synaptic and post-synaptic neurons increases due to the sensitization. This enhanced synaptic strength activates the post-synaptic motor neuron. Consequently, when a touch stimulus is perceived after sensitization, the motor neuron readily generates output spikes to the muscle fibers, inducing locomotion.

If a pain stimulus is perceived solely without a touch stimulus, the synaptic strength between pre-synaptic and post-synaptic neurons remains unchanged, remaining at a low level. This is because the pain is not associated with any tactile stimulation. In this case, the motor neuron does not generate output spikes in response to touch stimulus[7].

Based on the fabricated third-order $HfO_2$ memristor, a MASNS with habituation and sensitization characteristics was developed (see Fig. 4b). The responses of the developed MASNS to the various stimulus conditions were examined as shown in Figs. 4c, d. In the developed MASNS, a tactile neuron module and a pain (electric shock) sensor module were employed to sense touch and pain stimuli, respectively, mimicking a biological SNS (see Fig. 4b). Upon perception of a touch or pain stimulus by the sensors, a pulse generator module transfers specific voltage pulses followed by a read pulse to the memristor. The read pulse generates an output current through the memristor, and the motor module induces a response (rotating) when the output current exceeds a specific threshold.

In the developed MASNS, a positive voltage pulse is applied to the third-order memristor upon sensing a tactile stimulus (touch), while a negative voltage pulse is applied upon sensing a pain stimulus (electric shock), as shown in Fig. 4b, c. There are three different input pulse scenarios depending on the applied stimuli (see Fig. 4c). When the touch stimulus is repeatedly applied without electric shock, the device conductance initially increases but soon decreases due to the habituation characteristic of the third-order memristor (see Fig. 4d). On the other hand, when the electric shock is repeatedly applied, the negative voltage pulse for the pain stimulus reduces the device conductance, maintaining it at a high resistance state. If both touch and electric shock stimuli are applied together, a combined voltage pulse with the negative and positive voltage pulses is applied to the third-order memristor. The combined voltage pulses adapt the device into low resistance state over repeated touch and electric shock, regardless of the pulse order (see Supplementary Fig. S23). The potentiation of the device in response to touch and electric shock indicates a strong synaptic connection between the pre-synaptic tactile neuron and the post-synaptic motor neuron. The synaptic characteristics, such as the number of stimuli required for sensitization or degree of dishabituation against a threatening event, can be finely adjusted by modifying the pulse condition for each stimulus, as shown in Supplementary Figs. S24 and S25, respectively. The realization of habituation and sensitization characteristics in the device without the need of

bulky peripheral circuitries indicates the superior energy-efficiency of the third-order memristor-based MASNS for robotic applications. Supplementary Table S3 highlights the energy-efficiency of the third-order memristor-based MASNS compared to conventional circuit-based systems.

### Habituation and sensitization of a robot arm with MASNS

Implementing MASNS in robotic systems could improve energy efficiency while reducing processor burden by effectively filtering out insignificant external signals, as shown in Supplementary Fig. S26. To investigate the effectiveness of the third-order memristor for an intelligent MASNS, we built a MASNS-implemented robot arm system (see Supplementary Fig. S27 and "Methods"). The robot arm generates movements in response to external stimuli that produce a memristor output current exceeding a specific threshold, analogous to the response of a motor system to sensory stimulation that generates excitatory post-synaptic potentiation exceeding a threshold potential in biological SNS (see Supplementary Fig. S28). The reaction of the robot arm against external stimuli (touching and electric shock) was tested with both the low-order and the third-order memristors (see Fig. 5).

In the experiment, the robot arm was stimulated five times by touching a pen. Subsequently, touching and electric shock stimuli were simultaneously applied by touching the robot arm with a 5 V charged metal tip in the 6th event, followed by three instances of touch stimuli in the 7th to 9th events. In the case of the MASNS-implemented robot arm with the low-order memristor (5 nm of $HfO_2$ layer), the conductance of the memristor monotonically increased as the touch stimuli were applied due to the monotonous conductance update of the low-order memristor (see Fig. 5b). The robot arm's response after applying stimulus at the 1st, 3rd, 5th, and 6th events are shown in Fig. 5d. As shown in Fig. 5b, d, and Supplementary Movie 1, the robot arm generated rotating locomotion, which corresponds to the avoidance action to the dangerous signals, for every safe touch stimulus as well as for the threatening electric shock at the 6th event. In other words, the robot arm with the low-order memristor-based MASNS wasted energy and time by sensitively responding to insignificant and safe stimuli.

On the other hand, the robot arm with the third-order memristor-based MASNS was tested in the same manner, and the results are shown in Fig. 5c, e, and Supplementary Movie 2. Due to the habituation characteristics of the developed third-order memristor, the conductance of the memristor did not monotonically increase with the application of voltage pulses. The conductance of the memristor abruptly increased when the first touch stimulus was applied, and the robot arm generated a strong response against the first touch stimulus due to the strong synaptic strength in the MASNS (see Fig.5c, e). However, as the safe touch stimuli were consecutively applied, the conductance of the memristor decreased, and the response of the robot arm was also gradually reduced. Consequently, the robot arm did not react against the safe touch stimulus after the 4th stimulus was applied, which means that the robot arm does not consume energy and time to respond to the insignificant external stimuli.

While habituation makes an animal ignore insignificant and safe stimuli, the sensitization of the biological SNS enables an animal to sensitively respond to a threatening stimulus (or a stimulus that was once safe but is now threatening). Without sensitization, recognizing stimuli that were once safe but are now threatening becomes difficult. Therefore, sensitization is an essential function for avoiding threats. As shown in Fig. 5c, e, the conductance of the third-order memristor, habituated at the 5th event, abruptly increased when the voltage pulse for the "touching + electric shock" case in Fig. 4d was applied at the 6th event. Then, the robot arm with the third-order memristor-based MASNS generated a strong response at the 6th event. The robot arm sensitively responded to the following safe stimulus at the 7th and 8th

touching events until habituation occurs, demonstrating similarity to the sensitization of the biological SNS.

To compare the responses of each MASNS under severe stimuli, conductances of the low-order memristor and the third-order memristor were measured by applying 20 cycles of consecutive 24 touch stimuli followed by a single touching and electric shock stimulus (see Fig. 5f, g, and Supplementary Fig. S29). The applied stimuli contain a total of 480 safe touch stimuli, and the number of events that the conductance exceeds the threshold was counted. As shown in Fig. 5f and Supplementary Fig. S29a, the low-order memristor's conductance exceeded the threshold for 479 events among 480 events, demonstrating that a robotic system with the low-order memristor-based MASNS will consume unnecessary energy to respond to the safe and insignificant stimulus. The third-order memristor's conductance, however, did not exceed the threshold for 341 events among 480 events due to habituation, effectively ignoring approximately 71% of safe and insignificant stimuli (see Fig. 5g and Supplementary Fig. S29b). At the same time, the conductance of the third-order memristor exceeded the threshold for every electric shock stimulus due to sensitization. It is noted that long-term habituation, a decrement of overall synaptic strength according to repetitive input stimuli in the biological SNS, is observed in third-order memristor, as shown in Fig. S29b[42]. The results prove that the third-order memristor-based MASNS prevents unnecessary energy consumption in robotic systems while allowing sensitive responses to threats through habituation and sensitization, similar to the biological SNS.

## Discussion

In this study, a robot arm system with habituation and sensitization is experimentally demonstrated by implementing the third-order memristor-based artificial sensory nervous system (MASNS). A high-performance $HfO_2$-based third-order memristor was fabricated on an 8-inch wafer. The developed third-order memristor possesses three state variables including the filament width, device temperature, and the $TiO_x$ layer conductance, whereas conventional memristor devices have one or two state variables. The third-order memristor exhibits complex resistive switching behavior including habituation, sensitization, and frequency-dependent plasticity, owing to its three state variables. By connecting touch and pain (electric shock) sensor modules with a pulse generator, the third-order memristor successfully mimics the synaptic behaviors of the biological SNS with habituation and sensitization. Furthermore, the low-order and third-order memristor-based MASNS were directly implemented into the robot arm to analyze its response to safe or threatening stimuli. The third-order memristor-based MASNS enables the robot arm to ignore 71% of safe touch stimuli while sensitively responding to every threatening electric shock, demonstrating the effect of habituation and sensitization in terms of energy saving and avoiding threats. The developed third-order memristor and the MASNS present a promising approach for future intelligent robotic systems capable of efficiently handling external stimuli based on experiences without increasing processor loads or latency.

## Methods

### Device fabrication

The fabricated third-order memristor has a TiN/Ti/$HfO_2$/TiN structure. The 10 nm of $HfO_2$ switching layer (5 nm for the low-order memristor) is deposited through atomic layer deposition (ALD) system on the sputtered TiN bottom electrode. The 10 nm of Ti layer and the 150 nm of TiN layer are sequentially sputtered on the $HfO_2$ layer, and the active region is defined by dry etching with $SF_6$ gas. After defining the active region, the device is passivated with $SiO_2$ and the metal via and wires are fabricated through the back-end-of-line (BEOL) process. The maximum fabrication temperature of the BEOL process in this study is 400 °C, which is high enough to oxidize the Ti layer into the $TiO_x$ layer

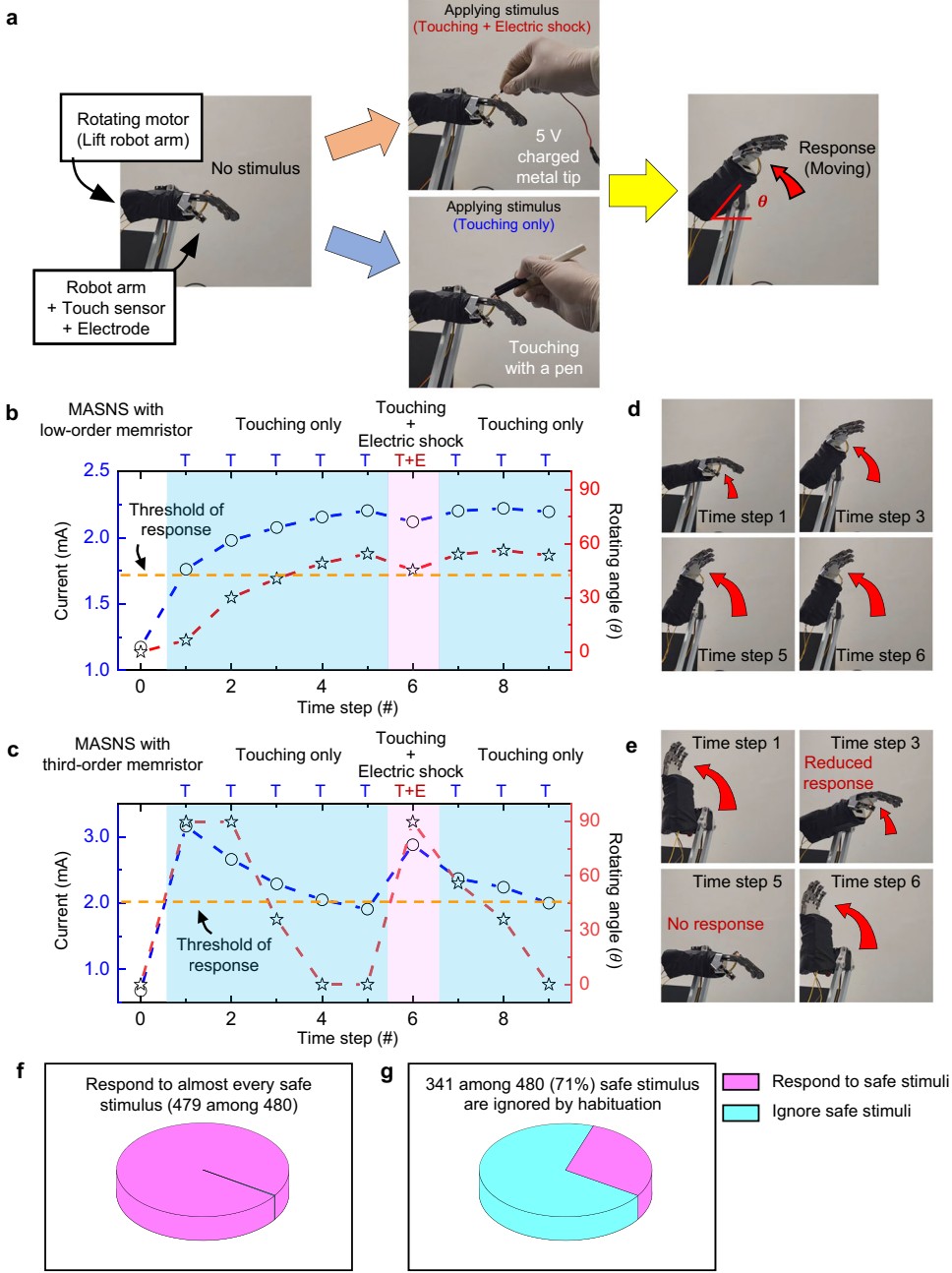

**Fig. 5 | MASNS-implemented robot arm having habituation and sensitization characteristics. a** Image of the MASNS-implemented robot arm system. Two types of stimuli ("touching + electric shock" or "touching only") are randomly applied to the robot arm, and the robot arm generates a rotation response according to the conductance of the third-order memristor. Conductance changes of (**b**) the low-order or (**c**) third-order memristor in the MASNS-implemented robot arm (blue dotted line) and the corresponding rotating angles of the robot arm (red dotted line) for the applied stimuli. **d**, **e**. Images of the robot arm at time steps 1, 3, 5, and 6 in (**b**, **c**), respectively. The third-order memristor-based MASNS-implemented robot arm ignores safe stimulus at time step 5, but it sensitively reacts to the threatening stimulus at time step 6. Results of the low-order (**f**) or third-order (**g**) memristor-based MASNS implemented robot arms responses against 480 safe stimuli.

by attracting oxygens from the $HfO_2$ layer. The final device stack after the fabrication is $TiN/Ti/TiO_x/HfO_2/TiN$, as illustrated in Fig. 2a–c. It is noteworthy that potential issues may arise during large-scale array integration of the device, such as *IR*-drop due to the high device conductivity and sneak-path problems within the array. Material optimization to reduce the overall device conductivity, alongside the integration of appropriate selector devices, could mitigate these issues and facilitate the realization of large-scale integrated third-order memristor arrays.

## Electrical characteristics measurement

The I-V sweep results of the fabricated device are measured through Keithley 4200A-SCS, by sweeping the voltage to the top electrode while grounding the bottom electrode and sensing the output current. The conductance update curves of the device according to the input voltage pulses are measured through a data acquisition tool (DAQ) and a current preamplifier. The DAQ generates a voltage pulse and measures the amplified output current through the device and the preamplifier.

### Third-order memristor modelling

The conductance update characteristics of the third-order memristor were simulated using MATLAB by modelling two anti-serially connected resistive switching variables: the filament diameter and $TiO_x$ layer conductivity. Initially, the conductance of the filament is set to be low (1.9 mS), while that of the $TiO_x$ layer is set to be high (9.2 mS). Then, consecutive 100 set pulses are applied, which potentiate or depress the conductance of the filament or $TiO_x$ layer, respectively. For the cases with short set pulse widths (1 and 2 μs), the resistive switching in the $TiO_x$ layer is negligible, resulting in a monotonic increment in the conductance update curves. However, in the longer set pulse width cases (3 and 4 μs), the conductance depression in the $TiO_x$ layer is dominant, leading to non-monotonic conductance update curves with consecutive set pulses. After applying set pulses, consecutive 100 reset pulses are applied to the device, potentiating the conductance of the $TiO_x$ layer while lowering the filament's conductance. It is noted that the voltage division between the filament and $TiO_x$ layer is considered in the simulation.

### Robot arm experiments

The robot arm system with MASNS was built and experimentally tested to investigate the effectiveness of the third-order memristor-based MASNS in robotic applications. The composition and the operation method of the manufactured robot arm system are as follows; a commercialized pressure (tactile) sensor and an electrode, which are connected to the controller (Arduino Uno), are attached to the robot arm to sense a tactile stimulus and an electric shock, respectively. A rotating motor is connected to the robot arm to generate locomotion (lifting the arm). When the robot arm is touched with a pen, the tactile sensor transfers the touch stimulus to the controller, and the DAQ applies a voltage pulse for the "touching only" case in Fig. 4c to the memristor. When the robot arm is touched with a 5 V charged metal tip (touching and electric shock are applied together), the touch sensor and the electrode perceive the touching and electric shock stimulus, respectively. Then, the DAQ applies a voltage pulse for the "touching +electric shock" case in Fig. 4c to the memristor. The pulse condition for each case is explained in Supplementary Fig. S29. After applying a voltage pulse to the memristor, the conductance of the memristor is measured by the following read voltage pulse. The rotating angle of the robot arm is determined by the conductance of the memristor. If the device conductance does not exceed the threshold, the rotating motor does not generate locomotion. On the other hand, if the device conductance exceeds the threshold, the controller operates the motor and rotates the robot arm. The rotating angle is decided by the device conductance (the detailed relationship between the memristor's conductance and the rotating angle is described in Supplementary Fig. S27).

## Data availability

The data required for assessing the conclusions are provided in the Source Data file. Source data are provided with this paper.

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

## Acknowledgements

This work was supported by National R&D Program through the National Research Foundation of Korea (NRF) funded by the Korean government (MSIT) (2022M3I7A2078273, 2022M3F3A2A01072851, 2020R1C1C1007464, RS-2024-00401234, RS-2025-00555433, and RS-2025-04162969), and Nanomedical Devices Development Project of NNFC. The EDA tool was supported by the IC Design Education Center (IDEC), Korea.

## Author contributions

S.P. and S.C. conceived this work. S.P. and H.J. designed the robot arm experiment. S.P., H.J. and S.S. measured the device and analyzed the device switching mechanisms. S.P., Y.K. and J.L. fabricated the device and conducted TEM imaging. S.P. and H.J. prepared the experiment movies. S.P. and S.C. prepared the manuscript. J.L. and S.C. supervised the study.

## Competing interests

The authors declare no competing interests.
