## [Transparent Peer Review file · Nature Communications]

Experimental Demonstration of Third-Order Memristor-based Artificial Sensory Nervous System for Neuro-Inspired Robotics

Corresponding Author: Professor Shinhyun Choi

Version 0:

Reviewer comments:

Reviewer #1

(Remarks to the Author)

This manuscript presents the development of a third-order memristor with a TiN/Ti/TiO_x/HfO₂/TiN structure. This memristor incorporates filament diameter, device temperature, and TiO_x conductivity as state variables, enabling it to emulate complex synaptic behaviors such as habituation and sensitization found in biological sensory nervous systems. By integrating this third-order memristor into a memristor-based artificial sensory nervous system (MASNS), the authors demonstrate its application in a robotic arm system. The robotic arm equipped with the MASNS effectively ignores approximately 71% of safe and familiar stimuli (habituation) while sensitively responding to threatening stimuli (sensitization).

This work is quite interesting; substantial efforts have been made across multiple layers, including device fabrication, characteristic modeling, and the robotic arm system. However, the following issues need to be addressed:

1. The authors need to make a detailed comparison between their device and other memristor devices that can simulate habituation (e.g., the following works), to highlight the advancements of their research. From these studies, it seems that achieving habituation functionality does not necessarily require a third-order memristor.

(1) Z. Wu, et al., "A Habituation Sensory Nervous System with Memristors," *Advanced Materials*, 2020.

(2) X. Li, et al., "Implementation of habituation on single ferroelectric memristor," *Applied Physics Letters*, 2023.

(3) R. Jiang, et al., "Habituation/Fatigue behavior of a synapse memristor based on IGZO–HfO₂ thin film," *Scientific Reports*, 2017.

2. In terms of device modeling, when the pulse reaches 4 μ S, the experimental data shows that it takes only two pulses to rise to the highest potential, whereas the modeling results indicate many more pulses are required. How do the authors explain this discrepancy between the experimental data and the model?

3. Figures 4-5 and the video demonstrate the device's performance under a combination of tactile and electrical stimuli. If these two stimuli are applied in reverse order, what would be the test results of the device (which is likely to occur in practical situations)? Would the habituation and sensitization emphasized in this paper still hold under these conditions?

4. Although the supplementary materials provide the endurance characteristics of the device, the authors should perform a more detailed electrical characterization of the designed device, such as retention measurements and other relevant parameters. These characteristics are crucial for the sensing applications of the robotic arm. Do the current characteristics meet the application requirements in such scenarios? The authors are encouraged to further analyze the relationship between application requirements and device characteristics to determine the direction and objectives for device optimization.

5. What is the yield of the proposed device? Is there potential for integrating such individual devices into sensor arrays, which is necessary for future applications?

6. Do the device characteristics exhibit significant changes with varying environmental temperatures? Assessing the stability and reliability of the device under different temperature conditions is important for practical applications.

7. While the authors have provided some explanations in the supplementary materials and the main text, the process of integrating the device into the robotic arm, particularly the conversion of tactile/electrical stimuli into electrical signals, needs to be further described. A detailed account of the signal transduction pathway and the interfacing mechanisms would enhance the understanding of the system's operation.

Reviewer #2

(Remarks to the Author)

The authors use a combination of two layers namely HfO₂ and TiO_x to modify combined electrical properties as function of voltage pulses to demonstrate different rates of responses (they refer to as learning). The results are used to control the response of a robot arm by different extents. Technical comments on the manuscript:

- Please refer to habituation and sensitization terminology commonly used in neuroscience to demonstrate the equivalent responses here. It is not clear how the authors show absence of fatigue for instance, dishabituation etc. which are all basic measurements needed to support the argument for single stimulus based learning.
- The learning should be dependent on the stimulus strength, I do not see this discussed, please clarify.
- Sensitization memory timescale is dependent on the stimulus strength. This is not shown anywhere.
- Typical habituation involves forgetting as the stimulus is presented at different intervals. In this case, if there is no change in memory, then I am not sure this is habituation, but rather just a drop in the strength of the e-field across the layer that is changing the property (e.g conductance).
- It is entirely not clear what the purpose of the robot arm is in this study, since the motion of this arm seems simply dependent on the strength of the stimulus. Is there a new capability that arises? Isn't the response obvious in terms of less signal strength in, less response amplitude out?
- So this is simply saturation or self-limiting due to reduced voltage being dropped in the dielectric, rather than habituation. In which case, it is not clear if much of the discussion in the Introduction etc is relevant to the paper.
- Please discuss prior literature on habituation / sensitization experiments using memory devices and replication of early neuroscience studies including use of identical material as in the current manuscript:
Zuo, F, et al. "Habituation based synaptic plasticity and organismic learning in a quantum perovskite." *Nature communications* 8.1 (2017): 240.
Jiang, R, et al. "Habituation/Fatigue behavior of a synapse memristor based on IGZO–HfO₂ thin film." *Scientific Reports* 7.1 (2017): 9354.
Mondal, S, et al. "All-electric nonassociative learning in nickel oxide." *Advanced Intelligent Systems* 4.10 (2022): 2200069.
Yang, X, et al. "Nonassociative learning implementation by a single memristor-based multi-terminal synaptic device." *Nanoscale* 8.45 (2016): 18897-18904.
- The authors present a Methods section on modeling of the device, but it is not clear how this is relevant to the paper. Is the model used instead of the device to move the robot? Since there exist numerous papers already on both TiO_x and HfO₂, is this model providing any new information or validates some previous hypothesis etc?
- The authors claim the memristor response is equivalent to the sea slug's behavior, but this is clearly wrong, since there is no timescale involved in the device due to the non-volatile nature. Hence, it is entirely not clear if the authors have mis-interpreted the results of effective low field dropping across the O migration layer being mistaken for non-associative learning.

Overall, given the numerous literature on habituation learning (including with interval training) in various flavors of memristor devices including the exact material reported in this study, namely HfO₂, it is far from obvious what is the novel result or scientific insight in this manuscript.

Further, given the electrical data does not contain temporal information, it is questionable whether the results should be considered in the context of 'habituation' that is borrowed from neuroscience or simply due to reduced current or voltage being applied to the second layer.

Reviewer #3

(Remarks to the Author)

The authors experimentally demonstrate a third-order memristor-based artificial sensory nervous system for neuro-inspired robotics, which is an interesting work. However, the following questions should be addressed before the publication.

1. The author attributes the effect of longer pulse time entirely to the change in temperature caused by Joule heat. Is there any evidence for this? I agree that higher temperatures would speed up the ion migration. But the repeated pulses with longer pulse time can also enhance the ion migration even if the temperature is fixed especially in the HRS state.
2. It should be noted that the device at a high resistance state should act as a capacitor. When the device changes from HRS to LRS, there is possibly a capacitor discharge current in addition to conducting current, which may lead to "overshoot" current (like habituation behavior). This should be verified.
3. During the sensitization process, the number of pulses (experienced in 1st stage of Fig.S8(a)) is minimal, which is inconvenient to the application. Can the sensitization process be well controlled by more pulse numbers? It is better to demonstrate this by adjusting pulse off time or amplitude.

4. What is the initial state in the TiOx? As shown in Fig.3(c), the conductance change in the TiOx layer is very small. Can the device work if no TiOx layer in the device? The switching mechanism of the device is still unclear.

Version 1:

Reviewer comments:

Reviewer #1

(Remarks to the Author)

I appreciate the authors' thorough responses to my previous queries. The additional experiments and analyses have significantly strengthened the manuscript.

Two areas still require attention:

1. While MASNS is claimed to improve energy efficiency, quantitative energy consumption data is lacking. Please provide measurements of the third-order memristor's power requirements across different operation modes (habituation, sensitization, standby) compared to conventional approaches.
2. The manuscript demonstrates excellent individual device performance but insufficiently addresses large-scale integration challenges. Please discuss potential issues (crosstalk, thermal effects, etc.) when integrating multiple devices and propose mitigation strategies.

Reviewer #2

(Remarks to the Author)

The authors have performed several additional experiments and provided clarifications. In response to one of the main questions concerning the analog between biology and their device, the authors state weak resetting pulses provide a pseudo-form of relaxation, since this is the closest they can come to in a non-volatile device. I assume this is a trade off between using a NVM device for memory versus synaptic learning and cannot be reconciled further with this device design. If the authors choose to, this may be worth discussing in the revised manuscript. I do not have further comments.

Response Letter to Reviewers' Comments

We sincerely appreciate the valuable time the reviewers have spent reviewing our manuscript and providing insightful comments and suggestions to help further improve the quality of our work. In response to the reviewers' evaluations, we have made a point-by-point response to the reviewers' comments and appended additional experimental results in our revised manuscript. We have also enhanced the clarity of the manuscript to better convey our findings. We believe we have addressed all of the reviewers' comments, and now the paper is more rigorous in content and clearer in presentation. Based on the responses below, we have updated one figure in the revised manuscript, added 17 supplementary figures, one supplementary table, and one supplementary note in the revised Supplementary Information, to address the comments. Our detailed point-by-point responses to the reviewers' comments are as follows.

Reviewer #1

This manuscript presents the development of a third-order memristor with a TiN/Ti/TiO_x/HfO₂/TiN structure. This memristor incorporates filament diameter, device temperature, and TiO_x conductivity as state variables, enabling it to emulate complex synaptic behaviors such as habituation and sensitization found in biological sensory nervous systems. By integrating this third-order memristor into a memristor-based artificial sensory nervous system (MASNS), the authors demonstrate its application in a robotic arm system. The robotic arm equipped with the MASNS effectively ignores approximately 71% of safe and familiar stimuli (habituation) while sensitively responding to threatening stimuli (sensitization).

This work is quite interesting; substantial efforts have been made across multiple layers, including device fabrication, characteristic modeling, and the robotic arm system. However, the following issues need to be addressed:

Response: We sincerely appreciate the reviewer's constructive comments on our work and valuable suggestions for enhancing the quality of the study. As the reviewer suggested, we have compared the characteristics of the proposed device with those of several devices emulating habituation, revised the device model, and clarified the detailed operation algorithms and signal pathway of the robot arm system. Additionally, we measured various electrical characteristics of the device including retention, switching speeds, uniformity, and yield to evaluate the suitability of the device for reliable robotic applications. We have also investigated the thermal stability and temperature effects of the device, demonstrating its ability to operate at elevated temperatures. These measured device performances indicate that the device possesses favorable characteristics for large-scale, reliable robotic applications in diverse environments. Our detailed responses to the reviewer's comments are provided below.

Comment #1:

The authors need to make a detailed comparison between their device and other memristor devices that can simulate habituation (e.g., the following works), to highlight the advancements of their research. From these studies, it seems that achieving habituation functionality does not necessarily require a third-order memristor.

(1) Z. Wu, et al., "A Habituation Sensory Nervous System with Memristors," *Advanced Materials*, 2020.

(2) X. Li, et al., "Implementation of habituation on single ferroelectric memristor," *Applied Physics Letters*, 2023.

(3) R. Jiang, et al., "Habituation/Fatigue behavior of a synapse memristor based on IGZO–HfO₂ thin film," *Scientific Reports*, 2017.

Response: We deeply appreciate the reviewer for this constructive comment. As the reviewer mentioned, some memristor devices have been reported to emulate synaptic properties such as habituation and sensitization. Therefore, a detailed comparison is essential to highlight the advancements of the proposed device. To address the reviewer's comments, we have compared various characteristics of the proposed device with those of previously reported memristor devices exhibiting habituation. In addition, we have also discussed the necessity of the third-order memristor to achieve both a non-volatile habituation state and the emulation of various synaptic properties simultaneously for a memristor-based artificial sensory nervous system (MASNS). The detailed explanations and results are shown below:

1) Comparison to other memristor devices for habituation

We have compared the proposed third-order memristor to other memristor devices that emulate habituation characteristics in a single device. The results of these comparisons are shown in Table R1. As summarized in Table R1, several devices have been developed to emulate habituation characteristics and memristor-based artificial sensory nervous systems (MASNS).

To enable MASNS for robotic applications, a memristor should satisfy various metrics, including CMOS-compatibility, non-volatile memory with long retention, good uniformity, fast switching speed, and habituation characteristics. However, previous studies have predominantly focused on mimicking biological synaptic properties rather than experimentally demonstrating robotic applications based on MASNS.

For instance, memristor devices based on Li_xSiO_y or LiTaO₃ have demonstrated biologically plausible habituation characteristics, including habituation, dishabituation, stimulus intensity dependence, stimulus frequency dependence, and spontaneous recovery^{1,2}. However, the lack of analysis regarding CMOS-compatibility and uniformity limits their utilization in reliable robotic applications.

Other studies have reported memristor devices using LiLaTiO (LLTO), SmNiO₃ (SNO), or NiO to emulate habituation characteristics³⁻⁵. However, these devices exhibited volatile memory with short retention, meaning that the trained information fades over time. Furthermore, their slow switching speeds (>0.4 s) impede the response time of MASNS.

Additionally, memristor devices utilizing HfO_x layers have been investigated for their CMOS-compatibility, which facilitates integration with current CMOS technology and Si substrates^{6,7}. However, while these devices support CMOS-compatible integration, they often exhibit monotonic conductance updates of low-order memristors, thereby lacking critical habituation characteristics such as dishabituation.

To highlight the advantages of the proposed third-order memristor, we measured several electrical performance metrics including retention, switching speeds, cycle-to-cycle uniformity, and device-to-device uniformity (Figure R1). The results showed that the proposed third-order memristor possesses non-volatile retention (10 years retention at 440 K), reasonable cycle-to-cycle uniformity (σ/μ of 0.039, 0.211 for LRS, HRS, respectively) and device-to-device uniformity (σ/μ of 0.07, 0.27 for LRS, HRS, respectively), and fast switching speeds (250 ns), as well as CMOS-compatibility and habituation characteristics. These favorable characteristics enable the developed memristor to experimentally demonstrate robotic applications based on MASNS.

To address the reviewer's comments and compare various devices emulating habituation, we have appended Table R1 and Figure R1 as Supplementary Table S1 and Supplementary Figure S5 in the revised Supplementary Information. The appended explanations in the revised manuscript are as follows:

Page 9, line 149: "The fabricated HfO_2 memristor exhibited favorable operation characteristics with **stable retention, fast switching speed**, low switching variations, and good endurance (Supplementary Fig. S5 and S6). **Furthermore, the high yield within 8-inch large-scale CMOS-compatible fabrication (Supplementary Figs. S7 and S8) and high thermal stability in elevated temperatures (Supplementary Fig. S9) enable integration of the device with sensor arrays and make the device suitable for reliable robotic applications in various environments.**"

Page 11, line 217: "**Detailed comparisons of the third-order memristor with several devices emulating habituation, as presented in Supplementary Table S1, highlight its favorable electrical characteristics for reliable robotic applications, as well as ability to mimic various synaptic behaviors.**"

Ref.	1	2	3	4	5	6	7	This work
Device	TiN/ Li _x SiO _y /Pt	Au/LiTaO ₃ /Pt	Pt/LLTO/ Pt	Pt/SNO/Pt	Pt/NiO/Pd	W/HfO _x / Ti/TiN	Al/HfO _x / IGZO/Au	TiN/HfO ₂ /TiO _x / Ti/TiN
CMOS Compatibility	No	No	No	No	No	Yes	No	Yes
Retention	10,000 s	>1,000 s	<200 s	12 h	20 s	N/A	N/A	10 years at 440.42 K
Cycle-to-cycle uniformity	N/A	N/A	N/A	N/A	2%	N/A	N/A	LRS: 3.9% HRS: 21.1%
Device-to- device uniformity	N/A	N/A	N/A	N/A	N/A	N/A	N/A	LRS: 7% HRS: 27%
Switching speed	100 ns	1 μs	0.4 s	200 s	0.5 s	1 ms	50 ms	250 ns
Habituation	Yes	Yes	Yes	Yes	Yes	Yes	Yes	Yes
Dishabituation	Yes	No	Yes	No	No	Yes	No	Yes
Habituation of dishabituation	Yes	No	Yes	No	No	No	No	Yes
Stimulus intensity dependence	Yes	Yes	Yes	No	Yes	Yes	Yes	Yes
Stimulus frequency dependence	Yes	Yes	Yes	No	Yes	Yes	Yes	Yes
Potentiation of habituation	Yes	No	No	No	No	No	No	Yes
Spontaneous recovery	Yes	No	Yes	Yes	Yes	No	Yes	Yes
Application	Robot navigation (Simul.)	N/A	N/A	Spiking neural network (Simul.)	Homeostatic learning (Simul.)	N/A	N/A	Neuro-inspired robot arm control (Exp.)

Table R1. Comparison with various memristors emulating habituation characteristics. To demonstrate advantages of the third-order memristor for emulating habituation characteristics, several key characteristics are compared. For utilizing a memristor-based artificial sensory nervous system (MASNS) in robotic applications, non-volatile memory, good uniformity, fast switching speed, as well as ability to emulate various habituation characteristics are required. The comparison reveals the third-order switching complexity enables non-volatile habituation state and various habituation characteristics, while CMOS-compatible HfO₂-based resistive switching offers good uniformity and fast switching speeds.

Figure R1. Favorable electrical characteristics of the third-order memristor for large-scale real-world robotic applications. **a.** Retention characteristics of the device showing stable and non-volatile memory with 10 years retention at approximately 440 K. The retention was extracted by monitoring fail time at elevated temperatures (543, 553, 565, and 573 K) and extrapolating the fitted results. The extracted activation energy was 0.97 eV, close to the previously reported activation energy of oxygen vacancies in HfO₂⁸. **b.** Fast switching speeds of the device. The device exhibited abrupt resistive switching and habituation for pulse widths longer than 500 ns. For pulse widths as short as 250 ns, the device displayed resistive switching without habituation. The set pulse amplitude was 0.85 V, and the device was measured by read pulses (0.3 V and 100 μ s). **c** and **d.** Cycle-to-cycle variation of the device. Consecutive 100 *I-V* sweeps were applied to the device (**c**). The device exhibited stable and uniform bipolar resistive switching, with a variation coefficient (σ/μ) of 0.039 and 0.211 for LRS and HRS, respectively (**d**). **e** and **f.** Device-to-device variation of the device. The *I-V* curves from randomly selected 10 devices were compared (**e**). The randomly selected devices exhibited similar electrical characteristics without severe device-to-device variation, with a variation coefficient of 0.07 and 0.27 for LRS and HRS, respectively (**f**).

2) Necessity of third-order switching complexity for MASNS

As the reviewer pointed out, it is indeed possible to emulate some of the habituation characteristics without third-order switching complexity. Various devices have demonstrated habituation characteristics¹⁻⁵, or have mimicked habituation using conductance depression or saturation of low-order memristors^{6,7}. However, achieving both the required electrical performance and complex habituation behaviors simultaneously is challenging without third-order switching complexity involving three state variables.

For instance, devices that exhibit habituation through mechanisms such as filament rupture driven by Joule heating¹ or ion diffusions³⁻⁵ typically display volatile habituation characteristics. The volatile memory of the habituation state causes the trained information to decay over time, posing significant challenges for reliable and stable real-world robotic applications. Conversely, devices emulating habituation through conductance depression or saturation of low-order memristors^{6,7} have primarily demonstrated only limited synaptic behaviors associated with habituation characteristics.

In contrast, the third-order memristor presented in our study achieves both a non-volatile habituation state and the emulation of various synaptic behaviors within a single device. The non-volatile resistive changes in the conductive filament and the inserted TiO_x layer ensure a stable habituation state (Figure R1a and Figure R1c). Furthermore, the third-order switching complexity enables the device to replicate various synaptic behaviors, highlighting its unique capability to fulfill the dual requirements of electrical performance and complex habituation behaviors for MASNS (Figure R2).

To emphasize the necessity of third-order characteristics for robotic applications with MASNS, we have included a detailed discussion on the advantages and limitations of previous studies employing low-order memristors in the revised manuscript. Additionally, we have appended Figure R2 in the revised Supplementary Information as Supplementary Figure S19, along with corresponding explanations to illustrate that the third-order memristor exhibits various synaptic behaviors related to habituation. We have also appended explanations regarding the necessity of third-order switching complexity for achieving stable habituation states while simultaneously satisfying various habituation characteristics. The appended explanations are as follows:

Page 11, line 212: “**Despite the absence of short-term memory, the third-order memristor demonstrated various synaptic behaviors associated with habituation, as shown in Supplementary Figs. S19, S20, and Supplementary Note S1. Furthermore, certain biological synaptic behaviors reliant on short-term memory, such as spontaneous recovery and potentiation of habituation, can be mimicked by periodically applying weak reset pulses, if necessary (see Supplementary Figure S19g, S19h, and S20). Detailed comparisons of the third-order memristor with several devices emulating habituation, as presented in Supplementary Table S1, highlight its favorable electrical characteristics for reliable robotic applications, as well as ability to mimic various synaptic behaviors.**”

Figure R2. Synaptic behaviors associated with habituation. **a.** Dishabitation. The 100 sets of positive voltage pulses (0.75 V and 5 μ s) were applied to the device, resulting in the habituation. After the habituation, consecutive 10 combined negative (-0.9 V and 1 μ s) and positive (0.75 V and 5 μ s) voltage pulses corresponding to pain and tactile stimuli, respectively, were applied to the device. The presentation of the combined pulses increased the device conductance back to LRS, indicating the dishabitation. **b.** Habituation of dishabitation. The 100 sets of positive voltage pulses (0.75 V and 5 μ s) were applied to the device for habituation. After the first habituation, consecutive 100 dishabitation combined voltage pulses (-0.9 V, 1 μ s and 0.75 V, 5 μ s) were applied to the habituated device. The device conductance initially increased by the dishabitation, but it soon decreased as the dishabitation pulses continued, indicating the habituation against the dishabitation stimuli. **c** and **d.** Stimulus frequency dependency. To investigate the effect of stimuli frequency, two positive voltage pulses (0.75 V, 3 μ s) with different pulse intervals (from 1 μ s to 10 ms) were repetitively applied to the device (**c**). The device exhibited the stimulus frequency dependency with faster habituation and lower final output current for shorter pulse intervals or higher frequency (**d**). **e** and **f.** Stimulus intensity dependency. Consecutive 100 positive set voltage pulses with different amplitudes of 0.75, 1.00, 1.05, 1.10 V and width of 5 μ s were applied to the device (**e**). The device exhibited pronounced habituation for the 0.75 V case. However, the higher pulse amplitude resulted in higher output currents, due to thicker filament formation (**f**). The results demonstrate the stimulus intensity dependency of the third-order memristor. **g.** Spontaneous recovery. Spontaneous recovery requires time-decaying memory or volatile memory. To emulate spontaneous recovery in the non-volatile third-order memristor, periodic weak reset pulses (-0.7 V, 2 μ s) were utilized. After 100 sets of positive voltage pulses (0.75 V, 5 μ s) for habituation, positive voltage pulses were withheld for 100 seconds. During this withholding period, the periodic weak reset

pulses were applied to the device every 10 seconds. Partial recovery of the habituated state was observed after the 100 seconds of withholding period, demonstrating the pseudo-spontaneous recovery. **h.** Potentiation of habituation. Two stages of habituation events were presented (0.75 V and 5 μ s), with the 100 seconds of stimulus withholding period. During the withholding period, the periodic weak reset pulses were applied for pseudo-spontaneous recovery. It was observed that the second habituation was faster than the first habituation, demonstrating the potentiation of habituation. Here, the device output current was measured by the read pulse (0.3 V and 100 μ s).

Comment #2:

In terms of device modeling, when the pulse reaches 4 μ S, the experimental data shows that it takes only two pulses to rise to the highest potential, whereas the modeling results indicate many more pulses are required. How do the authors explain this discrepancy between the experimental data and the model?

Response: We sincerely thank the reviewer for this valuable comment. The modeling in the original manuscript consisted of two anti-serially connected memristive components, which are filament conductivity and TiO_x conductivity, respectively. However, since Joule heating plays a significant role in the switching characteristics of the third-order memristor, it was challenging to strictly model the device using only these two anti-serially connected memristive components. For instance, the resistive switching in the filament and TiO_x layer is enhanced with longer pulse widths, during which the generated Joule heat remains for a longer duration compared to shorter pulse widths.

To better reflect the experimental conductance update curves of the fabricated device, we have updated the model by enhancing the effect of the pulse width on the resistive switching of the filament and TiO_x layer. With the revised model, we have demonstrated that the revised model now more closely aligns with the experimental results, as shown in Figure R3.

While the revised model still shows some discrepancies with the experimental results, we would like to emphasize that the primary purpose of the model is to support the understanding of the switching mechanism in the third-order memristor. The model demonstrates that the non-monotonic conductance update curves, which are difficult to obtain with existing memristor models, arise from the interaction between the two anti-serially connected resistive switching elements.

Figure R3. Previous and revised device modeling results. The previous device model exhibited discrepancies in the experimental data, which may confuse readers. To closely reflect the conductance update curve of the device, the model was revised by enhancing the effect of the pulse widths on the resistive switching in the filament and TiO_x layer. The revised model exhibited a more similar conductance update curve to the experimental data.

To provide the revised modeling results, we have updated Figure 3 with the revised modeling results in the revised manuscript. We have also revised the manuscript to clarify the purpose of the model in this study as follows:

Page 13, line 228: “It matches well with the experimental data of the third-order memristor. **The identical non-monotonic conductance update curve from the model, which is hardly acquired in conventional low-order memristors, demonstrates that the habituation characteristics originate from the anti-serially connected resistive switching filament and TiO_x layer.**”

Comment #3:

Figures 4-5 and the video demonstrate the device's performance under a combination of tactile and electrical stimuli. If these two stimuli are applied in reverse order, what would be the test results of the device (which is likely to occur in practical situations)? Would the habituation and sensitization emphasized in this paper still hold under these conditions?

Response: We genuinely appreciate the reviewer for this insightful comment. In the developed robot arm system, we designed the pre-synaptic spike from pain stimuli to precede that of tactile stimuli when both stimuli are applied simultaneously. However, as the reviewer pointed out, a MASNS should exhibit similar dishabituation behavior regardless of the order in which the tactile and pain stimuli are applied, in order to enable flexible robotic applications across various environments. To address this concern, we have verified that the developed MASNS exhibits similar dishabituation behavior in response to combined stimuli regardless of pulse order, due to its unique third-order switching characteristics.

To demonstrate the response of the third-order memristor to the order of tactile and pain stimuli, we applied two different dishabituation stimulus cases with different orders, as shown in Figure R4. In this experiment, a negative voltage pulse represents a pre-synaptic spike of the pain stimuli, while a positive voltage pulse represents a pre-synaptic spike of the tactile stimuli, equivalent to the pulse condition in Figures 4 and 5 in the original manuscript.

When a negative voltage pulse precedes a positive voltage pulse, the device exhibits dishabituation behavior with increased conductivity (Figure R4a). The negative voltage pulse relocates oxygen anions from the TiO_x layer to the filament, bringing the device to an intermediate resistance state between a high resistance state (HRS) and a low resistance state (LRS) (see Figure R4b). The subsequent positive voltage pulse potentiates the growth of the filament in the HfO_2 layer, resulting in dishabituation and the device transitioning to LRS.

Similarly, the third-order memristor exhibits dishabituation behavior when the tactile and pain stimuli are applied in reverse order (Figure R4c). When the positive voltage pulse precedes the negative voltage pulse, the positive voltage pulse relocates oxygen anions from the filament to TiO_x layer, resulting in a stronger habituation state (Figure R4d). The subsequent negative voltage pulse removes oxygen anions from the TiO_x layer, returning the device to the LRS and completing the dishabituation behavior.

Figure R4. Conductance increases according to dishabituation stimuli with different pulse orders. The effects of the stimulus order in dishabituation stimuli were investigated by applying combined voltage pulse with different pulse orders to the habituated device while measuring the conductance change. **a** and **b**. Response of the device when negative voltage pulse precedes. When the negative voltage pulse (-0.9 V , $1\ \mu\text{s}$) for pain stimulus precedes the positive voltage pulse (0.8 V , $1\ \mu\text{s}$) for tactile stimulus, dishabituation with increased conductance was observed (**a**). The negative voltage pulse moves oxygen anions in the TiO_x layer into the filament, partially rupturing the filament and resulting in the intermediate state between HRS and LRS. Conversely, the following positive voltage pulse removes oxygen anions in the filament, increasing the device conductance (**b**). **c** and **d**. Response of the device when positive voltage pulse precedes. When the positive voltage pulse for tactile stimulus precedes the negative voltage pulse for pain stimulus, dishabituation with increased conductance was observed (**c**), similar to the reverse order case. The positive voltage pulse moves oxygen anions into the TiO_x layer, making the device more habituated. The following negative voltage pulse relocates oxygen anions from TiO_x layer to the filament, thereby increasing the device conductance (**d**).

This dishabituation behavior regardless of the stimuli order is achieved by the third-order switching complexity. In various memristor devices, a reset pulse with negative voltage decreases device conductivity. On the contrary, in the third-order memristor having the separated habituation state from high and low resistance states, a negative voltage pulse could increase device conductivity (see Figure R5a). By relocating oxygen anions from the TiO_x layer to the filament, the negative voltage pulse initially switches the device from habituation state to the LRS, and then switches it to the high resistance state after continuing the negative voltage pulses (see Figure R5b).

Figure R5. Non-monotonic conductance updates in potentiation and depression. **a.** Conductance update curves with consecutive 100 positive voltage pulses (0.75 V, 5 μ s), followed by consecutive 100 negative voltage pulses (-0.8 V, 1 μ s). The non-monotonic conductance update properties were observed in the depression stage, as well as in the potentiation stage. **b.** Illustrations about device switching mechanisms for each state.

To clarify the dishabituation behavior regardless of the stimuli order, we have appended the Figure R4 and R5 in the revised manuscript as Supplementary Figures S23 and S12, respectively. We have also modified the manuscript to include corresponding explanations as follows:

Page 15, line 284: “If both touch and electric shock stimuli are applied together, a combined voltage pulse with the negative and positive voltage pulses is applied to the third-order memristor. The combined voltage pulses adapt the device into low resistance state over repeated touch and electric shock, regardless of the pulse order (see Supplementary Fig. S23). The potentiation of the device in response to touch and electric shock indicates a strong synaptic connection between the pre-synaptic tactile neuron and the post-synaptic motor neuron.”

Page 10, line 178: “The device conductance state can be programmed back into the HRS by applying strong negative bias reset pulses or back into the LRS by applying weak negative bias reset pulses (see Fig. 2d and Supplementary Fig. S12).”

Comment #4:

Although the supplementary materials provide the endurance characteristics of the device, the authors should perform a more detailed electrical characterization of the designed device, such as retention measurements and other relevant parameters. These characteristics are crucial for the sensing applications of the robotic arm. Do the current characteristics meet the application requirements in such scenarios? The authors are encouraged to further analyze the relationship between application requirements and device characteristics to determine the direction and objectives for device optimization.

Response: We sincerely thank the reviewer for this constructive comment. As the reviewer pointed out, a more detailed electrical characterization is required to assess whether the device characteristics are suitable for robotic applications. Based on the reviewer's comment, we have conducted detailed characterizations of the device performance and its response to high temperatures. We have also revised the manuscript to clarify that the developed device possesses appropriate electrical characteristics for robotic applications. The detailed explanations and comments are shown below:

To utilize a memristor for robotic applications, a memristor-based synaptic device should satisfy various performance standards. For instance, a synaptic device should retain trained information over a long period without loss or decay, making non-volatile memory characteristics with long retention a preferred feature. In addition, a short switching time of a few microseconds is required for the fast response of robotic systems to external stimuli. Furthermore, good cycle-to-cycle uniformity and device-to-device uniformity are essential for ensuring reliable real-world robotic applications.

We measured various electrical characteristics, including retention, switching speeds, cycle-to-cycle uniformity, and device-to-device uniformity. The retention of the device was assessed by monitoring fail time at elevated temperatures (543, 553, 565, and 573 K) and extrapolating the fitted results using the Arrhenius plot (see Figure R6a). The fail time was defined as the time it takes for the output current to change by more than 50 μA . The extracted activation energy was 0.97 eV, which is close to the previously reported activation energy of oxygen vacancies in HfO_2 ⁸. The results show that the developed device possesses non-volatile memory with 10 years of retention at a high temperature of 440.42 K.

The switching speeds of the device were analyzed with voltage pulses of various pulse widths of 250, 500, 750, and 1,000 ns with a fixed voltage of 0.85 V, as shown in Figure R6b. Longer pulse widths than 500 ns generated abrupt resistive switching and fast habituation, while a short pulse width of 250 ns generated gradual resistive switching and no habituation. The results demonstrated that the developed device exhibits fast switching speeds of approximately 250 ns but requires longer pulse widths (greater than 500 ns) to exhibit habituation characteristics.

The cycle-to-cycle and device-to-device variations of the developed device were analyzed from consecutive 100 *I-V* sweeps or randomly selected 10 devices, respectively. As shown in Figure R6c and R6d, the device showed similar *I-V* curves for consecutive 100 cycles, achieving low cycle-to-cycle variation coefficients (σ/μ) of 0.039 and 0.211 at low resistance state and high resistance state, respectively. The device also exhibited low device-to-device variations, when

analyzed from randomly selected 10 devices in a single die (see Figure R6e and R6f). The measured I - V curves showed low variation coefficients of 0.07 and 0.27 at low resistance state and high resistance state, respectively. The good uniformity with low cycle-to-cycle and device-to-device variations demonstrates that the developed device could be a suitable option for reliable robotic applications.

Figure R6. Favorable electrical characteristics of the third-order memristor for large-scale real-world robotic applications. **a.** Retention characteristics of the device showing stable and non-volatile memory with 10 years retention at approximately 440 K. The retention was extracted by monitoring fail time at elevated temperatures (543, 553, 565, and 573 K) and extrapolating the fitted results. The extracted activation energy was 0.97 eV, close to the previously reported activation energy of oxygen vacancies in HfO_2 .⁸ **b.** Fast switching speeds of the device. The device exhibited abrupt resistive switching and habituation for pulse widths longer than 500 ns. For pulse widths as short as 250 ns, the device displayed resistive switching without habituation. The set pulse amplitude was 0.85 V, and the device was measured by read pulses (0.3 V and 100 μs). **c** and **d.** Cycle-to-cycle variation of the device. Consecutive 100 I - V sweeps were applied to the device (**c**). The device exhibited stable and uniform bipolar resistive switching, with a variation coefficient (σ/μ) of 0.039 and 0.211 for LRS and HRS, respectively (**d**). **e** and **f.** Device-to-device variation of the device. The I - V curves from randomly selected 10 devices were compared (**e**). The randomly selected devices exhibited similar electrical characteristics without severe device-to-device variation, with a variation coefficient of 0.07 and 0.27 for LRS and HRS, respectively (**f**).

The results demonstrate that the developed device possesses favorable electrical characteristics for artificial sensory nervous systems in robotic applications, with long retention, fast switching speed, and good cycle-to-cycle and device-to-device uniformity. To clarify that the developed third-order memristor has suitable electrical characteristics for robotic applications, we have appended Figure R6 as Supplementary Figure S5 in the revised Supplementary Information. We have also appended corresponding explanations about the electrical characterization results in the revised manuscript as follows:

Page 9, line 149: “The fabricated HfO₂ memristor exhibited favorable operation characteristics with **stable retention, fast switching speed**, low switching variations, and good endurance (Supplementary Fig. S5 and S6). **Furthermore, the high yield within 8-inch large-scale CMOS-compatible fabrication (Supplementary Figs. S7 and S8) and high thermal stability in elevated temperatures (Supplementary Fig. S9) enable integration of the device with sensor arrays and make the device suitable for reliable robotic applications in various environments.**”

Comment #5:

What is the yield of the proposed device? Is there potential for integrating such individual devices into sensor arrays, which is necessary for future applications?

Response: We sincerely thank the reviewer for this constructive comment. As the reviewer mentioned, the yield of the device in large-scale fabrication is one of the most significant factors for realizing future robotic applications. To address the reviewer’s comment, we conducted an assessment of device yield by characterizing randomly selected 30 devices in an 8-inch wafer.

We utilized a foundry-level 8-inch CMOS fabrication process for device manufacturing. Based on this mature fabrication process, a high device yield was achieved, as shown in Figure R7. We measured 30 randomly selected devices from the 8-inch wafer. All devices successfully demonstrated forming operations with similar forming curves (see Figure R7a). The obtained forming voltages were approximately 4.7 V, with minimal variations (see Figure R7b). Furthermore, we analyzed the switching operation of these 30 devices after forming (see Figure R8). As shown in Figure R8, all 30 devices demonstrated similar resistive switching behavior.

The results confirm that the developed device possesses close to 100% yield. This high yield is likely attributed to the mature CMOS fabrication process with optimized fabrication conditions. With this high yield and fully CMOS-compatible fabrication process, the device is expected to offer advantages for large-area integration with sensor arrays. Leveraging this high yield and reliable large-scale CMOS-compatible fabrication process, the developed device is anticipated to enable the integration of sensor and synapse components, making it suitable for robotic applications with MASNS.

We have revised the Supplementary Information to incorporate the yield information of the device in Figure R7 and R8 as Supplementary Figure S7 and S8, respectively, along with corresponding explanations. We have also revised the manuscript to discuss the possibility of integration of the device with sensor arrays for future applications as follows:

Page 9, line 149: “The fabricated HfO₂ memristor exhibited favorable operation characteristics with **stable retention**, **fast switching speed**, low switching variations, and good endurance (Supplementary Fig. S5 and S6). **Furthermore, the high yield within 8-inch large-scale CMOS-compatible fabrication (Supplementary Figs. S6 and S7) and high thermal stability in elevated temperatures (Supplementary Fig. S8) enable integration of the device with sensor arrays and make the device suitable for reliable robotic applications in various environments.**”

Figure R7. Yield of the third-order memristor. a. Forming curves obtained from randomly selected 30 devices within the 8-inch wafer. **b.** Uniform forming voltage distribution of the randomly selected 30 devices, demonstrating great yield with high uniformity of the third-order memristor. The high yield of the device is expected to originate from the mature CMOS-compatible processes with optimized fabrication conditions.

Figure R8. I-V curves from randomly selected 30 devices within the 8-inch wafer after forming. All the device exhibited similar resistive switching, indicating the high yield of approximately 100%.

Comment #6:

Do the device characteristics exhibit significant changes with varying environmental temperatures? Assessing the stability and reliability of the device under different temperature conditions is important for practical applications.

Response: We are truly thankful to the reviewer for the suggestion regarding the assessment of thermal effects. As the reviewer commented, it is necessary to assess the effect of the temperature on device operations and stability for practical applications. To analyze the effect of temperature on the device, we measured the device's electrical characteristics at various temperatures from 300, 373, and 473 K, as shown in Figure R9 and R10.

The device was measured by voltage sweep at various temperatures to assess the resistive switching at elevated temperatures (see Figure R9a, R9c, and R9e). The measured *I-V* characteristics results revealed that the device exhibits similar resistive switching at elevated temperatures without noticeable difference.

The stability of the device at each resistance state of high resistance, low resistance, and habituation states was analyzed by applying a read voltage pulse and monitoring the device output current over 1,000 seconds at each temperature (Figure R9b, R9d, and R9f). The device exhibited stable resistance states up to 473 K, demonstrating the high stability of each memory state of the device. The results demonstrate the high thermal stability of the developed device, which is a favorable characteristic for real-world robotic applications^{9,10}.

Figure R9. Resistive switching and thermal stability in elevated temperatures. **a.** *I-V* curve of the device at room temperature. **b.** Conductance stability for each state of the device at room temperature. **c.** *I-V* curve of the device at an elevated temperature of 373 K. **d.** Conductance stability for each state of the device at 373 K. **e.** *I-V* curve of the device at an elevated temperature of 473 K. **f.** Conductance stability for each state of the device at 473 K, showing good thermal stability.

The pulsed responses of the device were measured at each temperature condition, while varying set voltage pulse amplitudes, as shown in Figure R10. Consecutive 100 positive voltage set pulses (0.7, 0.75, 0.8, and 0.85 V, 1 μ s) followed by 100 negative voltage reset pulses (-0.9 V, 1 μ s) were applied to the device, and the device conductance states were measured by read pulses of 0.3 V and 100 μ s. For every temperature condition, the device exhibited resistive switching according to the applied voltage pulses. Notably, more pronounced and fast habituation phenomenon could be observed at elevated temperature compared to the room temperature (300 K) case. The strong habituation at higher temperature is expected to originate from accelerated reaction of oxygen anions at higher temperature. The results show that the device could be operated by pulses at various temperatures.

Figure R10. Pulsed response of the device with consecutive 100 set pulses followed by 100 reset pulses while varying set voltages and temperatures. **a.** Pulsed response of the device at room temperature (300 K). Habituation characteristics were observed for the 0.8 V of set pulse amplitude. **b.** Pulsed response of the device in elevated temperatures of 373 K, showing pronounced habituation. **c.** Pulsed response of the device in elevated temperatures of 473 K, showing faster habituation at lower set pulse amplitudes (0.75 V), as well as pronounced habituation.

The tests at various temperatures revealed that the device could be operated at higher temperatures without significant changes, making it suitable for practical robotic applications. The device exhibited similar resistive switching characteristics at various temperature conditions. The high stability of each memory state was also confirmed up to 473 K, which makes the device suitable for robotic applications. The pulsed response results demonstrated that the device could be operated with fast voltage pulses at various temperatures. To clarify the high thermal stability and effects of the environmental temperatures on the device operation, we have appended Figures R9, and R10 as Supplementary Figures S9, and S18 in the revised Supplementary Information. We have also appended corresponding explanations in the revised manuscript as follows:

Page 9, line 149: “The fabricated HfO₂ memristor exhibited favorable operation characteristics with **stable retention, fast switching speed, low switching variations, and good endurance** (Supplementary Fig. S5 and S6). **Furthermore, the high yield within 8-inch large-scale CMOS-compatible fabrication (Supplementary Figs. S7 and S8) and high thermal stability in elevated temperatures (Supplementary Fig. S9) enable integration of the device with sensor arrays and make the device suitable for reliable robotic applications in various environments.**”

Page 11, line 203: “In addition, measurements at elevated temperatures revealed more pronounced and fast habituation characteristics, demonstrating the effects of the high temperature on resistive switching in the TiO_x layer and habituation characteristics (see Supplementary Fig. S18). The results support the conclusion that the device temperature acts as a critical state variable in the developed third-order memristor.”

Comment #7:

While the authors have provided some explanations in the supplementary materials and the main text, the process of integrating the device into the robotic arm, particularly the conversion of tactile/electrical stimuli into electrical signals, needs to be further described. A detailed account of the signal transduction pathway and the interfacing mechanisms would enhance the understanding of the system's operation.

Response: We sincerely appreciate the reviewer for this constructive comments. As the reviewer mentioned, it is crucial to provide detailed explanations about the robot arm system operation sequence and signal pathway to enhance the readability of the manuscript. The operation algorithm of the robot arm system is described in Figure R11, and the detailed sequence is as follows:

The robot arm system with the third-order memristor-based MASNS can be discriminated into four components; 1) robot arm with tactile/electric sensors and rotating motor, 2) Arduino controller for assessing stimuli signal from sensors and for operating motors, 3) host computer and data acquisition (DAQ) for controlling overall system, applying appropriate voltage pulses to memristor, and reading output current from memristor, 4) the third-order memristor for determining the response of the robot arm for applied stimuli.

The tactile and electric sensors were attached to the robot arm to sense applied stimuli. When tactile or electric (pain) stimulus is applied, the sensors generate output voltage which is received by the Arduino controller. Then, the Arduino controller sends a signal to the DAQ informing which type of stimuli (tactile, pain, or both) is applied. According to the type of applied stimuli, the DAQ applies the corresponding voltage pulse, which is a pre-synaptic spike in the biological counterpart, to the third-order memristor. The applied voltage pulse modulates the conductance of the memristor, thereby updating the synaptic strength properly. Next, the DAQ applies a read voltage pulse to the memristor to generate an output current, which is a post-synaptic spike in the biological counterpart. Finally, according to the amplitude of this output current and the threshold current, the host PC calculates the amplitude of the response and controls the Arduino controller to operate the rotate motor.

To further enhance the understanding of the system's operation, we have appended Figure R11 and corresponding explanations as Supplementary Figure S28 to provide a detailed operation sequence of the developed robot arm system. We have also revised the manuscript to provide explanations about the signal pathway of the developed robot arm system as follows:

Page 17, line 297: “Implementing MASNS in robotic systems could improve energy efficiency while reducing processor burden by effectively filtering out insignificant external signals, as

shown in Supplementary Fig. S26. To investigate the effectiveness of the third-order memristor for an intelligent MASNS, we built a MASNS-implemented robot arm system (see Supplementary Fig. S27 and Methods). The robot arm generates movements in response to external stimuli that produce a memristor output current exceeding a specific threshold, analogous to the response of a motor system to sensory stimulation that generates excitatory post-synaptic potentiation exceeding a threshold potential in biological SNS (see Supplementary Fig. S28). The reaction of the robot arm against external stimuli (touching and electric shock) was tested with both the low-order and the third-order memristors (see Fig. 5).”

Figure R11. Operation algorithm of the robot arm system. The tactile and electric sensors were attached to the robot arm to sense applied stimuli. When a tactile or electric (pain) stimulus is applied, the sensors generate output voltage which is received by the Arduino controller. Then, the Arduino controller sends a signal to the DAQ informing which type of stimuli (tactile, pain, or both) is applied. According to the type of applied stimuli, the DAQ applies the corresponding voltage pulse, which is a pre-synaptic spike in the biological counterpart, to the third-order memristor. The applied voltage pulse modulates the conductance of the memristor, thereby updating the synaptic strength properly. Next, the DAQ applies a read voltage pulse to the memristor to generate an output current, which is a post-synaptic spike in the biological counterpart. Finally, according to the amplitude of this output current and the threshold current, the host PC calculates the amplitude of the response and controls the Arduino controller to operate the rotate motor.

Reviewer #2

The authors use a combination of two layers namely HfO₂ and TiO_x to modify combined electrical properties as function of voltage pulses to demonstrate different rates of responses (they refer to as learning). The results are used to control the response of a robot arm by different extents. Technical comments on the manuscript:

Response: We sincerely appreciate the reviewer for providing constructive comments and suggestions. To address the reviewer's comments, we conducted comprehensive studies on various synaptic behaviors demonstrating the habituation characteristics of the device, including dishabituation, habituation of dishabituation, stimulus frequency dependency, stimulus intensity dependency, potentiation of habituation, and spontaneous recovery. Additionally, we compared the developed third-order memristor to previous studies that reported memristors emulating habituation behavior. We have also clarified the suitability of the device's characteristics for robotic applications, in comparison to conventional memristors.

Based on the reviewer's comments, we believe the additional experiments demonstrate the habituation characteristics of the third-order memristor and highlight the advantages of the third-order switching complexity compared to previous studies. The detailed responses to the comments are presented below:

Comment #1:

Please refer to habituation and sensitization terminology commonly used in neuroscience to demonstrate the equivalent responses here. It is not clear how the authors show absence of fatigue for instance, dishabituation etc. which are all basic measurements needed to support the argument for single stimulus based learning.

Response: We truly thank the reviewer for this valuable comment. As the reviewer pointed out, synaptic behaviors such as dishabituation and stimulus frequency/intensity dependency, which are essential for determining whether habituation has occurred, should be demonstrated¹¹. To showcase the habituation characteristics of the developed third-order memristor, we measured key synaptic behaviors related to habituation, including dishabituation, habituation of dishabituation, stimulus frequency dependency, stimulus intensity dependency, potentiation of habituation, and spontaneous recovery. The detailed explanations are provided below:

1) Memristor as an artificial synapse

Before explaining the results of additional experiments, we would like to emphasize that the memristor functions as an artificial synapse and exhibits useful habituation characteristics of biological synapses, making it suitable for robotic applications. In the biological nervous system, such as in sensory systems or neural networks in the brain, pre-synaptic neurons transmit spike signals to post-synaptic neurons, with synapses located between the axons of pre-synaptic neurons and the dendrites of post-synaptic neurons (see Figure R12a). The

synaptic strength between pre-synaptic and post-synaptic neurons modulates the amplitude of the transmitted signal. In artificial nervous systems, memristors have been widely studied as an artificial synapse due to their analog characteristics. Since the amplitude of electrical signals passing through a memristor is determined by its electrical conductivity, the electrical conductivity of the memristor is analogous to synaptic strength.

In a biological sensory nervous system with habituation characteristics, the activity of the post-synaptic motor neuron is influenced by the synaptic strength between the pre-synaptic sensory neuron and the post-synaptic motor neuron (see Figure R12b). The motor neuron activity, or the synaptic strength, exhibits a non-monotonic response to identical stimuli, enabling animals to ignore insignificant stimuli, as shown in Figure R12c. In this work, the developed third-order memristor exhibits a similar non-monotonic trend in response to identical voltage pulses, while conventional low-order memristors show only a monotonic increase or decrease in conductance (see Figure R12c). Based on this non-monotonic conductance update property, we demonstrated that the memristor-based artificial sensory nervous system (MASNS) using the third-order memristor can enhance the efficiency of robotic applications, rather than merely demonstrating the biological plausibility of the developed device.

As the reviewer pointed out, there are some discrepancies in synaptic behaviors and mechanisms between the memristor and biological synapses. For example, the developed device exhibits non-volatile memory characteristics, while biological synapses exhibit both short-term and long-term memory. Additionally, the speed of synaptic modulation is much faster in the developed device (a few hundred nanoseconds) compared to biological synapses (a few milliseconds). Eventhough there are some discrepancies, the purpose of the memristor is to mimic specific synaptic characteristics of biological synapses required for particular applications, rather than to realistically emulate all the synaptic characteristics of biological synapses.

Furthermore, the discrepancies could be advantageous for real-world robotic applications. For example, the non-volatile memory of the memristor could enhance the energy-efficiency of robotic applications by preventing the loss of trained information and avoiding re-training. Moreover, the faster speeds of the memristor can facilitate the rapid processing of artificial nervous system in response to external stimuli.

To compare the developed memristor with biological synapses, we measured various synaptic behaviors associated with habituation, explained the similarities/differences with biological synapses, and elaborated on how these differences could benefit neuro-inspired robotic applications. Additionally, we compared the synaptic behaviors of previously reported memristor devices with those of the third-order memristor to highlight the superior synaptic characteristics of the third-order memristor.

Figure R12. Illustrations of the functionality of memristors as artificial synapses. **a.** The conceptual illustration about how the memristor functions as an artificial synapse in the biological nervous system. **b.** Biological sensory-motor system with habituation, where the strength of the response is modulated by the synaptic strength. In this system, the response or synaptic strength exhibits initial sensitization followed by depression due to the habituation characteristics. **c.** Comparisons between the third-order memristor and conventional low-order memristors. The developed third-order memristor exhibits non-monotonic conductance update curves and therefore, can be utilized for an artificial synapse with habituation. On the contrary, conventional memristors show monotonic conductance update, which is hardly utilized for MASNS with habituation.

2) Synaptic behaviors regarding habituation characteristics

As the reviewer pointed out, it is essential to demonstrate specific synaptic behaviors that exhibit habituation characteristics. To address the reviewer's comments, we measured several synaptic behaviors associated with habituation, including dishabituation, habituation of dishabituation, stimulus frequency dependency, stimulus intensity dependency, spontaneous recovery, and potentiation of habituation, within the developed third-order memristor (see Figure R13). The results show that the third-order memristor exhibits a variety of synaptic behaviors related to habituation, confirming that the device possesses the key habituation characteristics. The detailed explanations about each synaptic behavior are described as follows:

1) Dishabituation

In biological sensory nervous systems, applying different stimuli leads to a recovery of the reduced response to the original state¹¹. This recovery from a habituated state to a sensitized state is called dishabituation. Dishabituation is one of the most significant synaptic behaviors that help distinguish whether a reduced response is due to habituation rather than motor neuron fatigue. To test the dishabituation characteristic of the third-order memristor, we first applied repetitive positive voltage pulses (0.75 V and 5 μ s) corresponding to tactile stimuli to the device, resulting in the habituation state (see Figure R13a). After the habituation, combined negative and positive voltage pulses (-0.9 V, 1 μ s and 0.75 V, 5 μ s) corresponding to pain and tactile stimuli, respectively, were applied to the device in the habituation state. Notably, the application of this different stimulus, or combined voltage pulse, caused the device conductance to a low-resistance state (LRS), mirroring the biological dishabituation phenomenon. These results demonstrate that the device exhibits dishabituation, with the conductance of the habituated device recovering upon the presentation of different stimuli.

2) Habituation of dishabituation

In biological sensory nervous systems, repeated application of dishabituating stimuli—which initially disrupts habituation—leads to a gradual reduction in response. This phenomenon is known as the habituation of dishabituation. To observe this habituation of dishabituation in the third-order memristor, we applied consecutive dishabituating stimuli of the combined voltage pulses after the habituation process, as shown in Figure R13b. Initially, the dishabituating pulses caused a significant increase in the device conductance, resulting in dishabituation. However, as these dishabituating pulses were repeatedly applied, the conductance of the device gradually decreased, demonstrating the habituation of dishabituation.

3) Stimulus frequency dependency

When the frequency of stimulation is higher, the more rapid and pronounced habituation is observed in the biological sensory nervous systems, and this phenomenon is called stimulus frequency dependency. To assess the effect of frequency on habituation in the third-order memristor, we applied two positive voltage set pulses with different pulse intervals, followed by a read pulse. This pulse train was repeated 50 times for each interval, and the resulting conductance update behavior was compared by analyzing the final output current (see Figure R13c). As shown in Figure R13d, the device exhibited faster habituation and lower final output current for shorter pulse interval cases. This frequency dependency originates from the thermal effect of the device, where a shorter pulse interval generates larger Joule heat with less heat dissipation. The observed variation in the final output current with different habituation behaviors demonstrates the stimulus frequency dependency of the developed device.

4) Stimulus intensity dependency

When the stimulus intensity is lower, the more rapid and pronounced habituation is observed in the biological sensory nervous systems, and this phenomenon is called stimulus intensity dependency. To examine the stimulus intensity dependency of the device, we applied consecutive 100 positive set voltage pulses with varying pulse amplitudes from 0.75 to 1.1 V, and measured the output current using read pulses (see Figure R13e). As shown in Figures R13e and R13f, the device exhibited pronounced habituation with a lower final output current

at a pulse amplitude of 0.75 V. However, when higher amplitude voltage pulses were applied, the increased voltage induced the formation of a thicker filament, resulting in a higher final output current. These results demonstrate that the device exhibits stimulus intensity dependency, similar to the biological counterpart.

5) *Spontaneous recovery*

Spontaneous recovery refers to the recovery of a response after a period of stimulus absence after habituation. In several previous studies, spontaneous recovery has been realized in volatile memory characteristics of devices, where trained information naturally fades due to the short retention of the memory. The non-volatile memory of the third-order memristor might be considered a discrepancy compared to biological synapses. While volatile memory can simulate spontaneous recovery, it is challenging to develop non-volatile or long-term memory of trained information in such volatile memory devices. In contrast, the non-volatile memory offers advantages in terms of reliability and energy-efficiency for robotic applications, as short-term memory devices would lose trained information over time or when powered off, making them unsuitable for such applications.

Furthermore, spontaneous recovery in non-volatile devices can be simulated by applying periodic weak reset pulses, which we termed pseudo spontaneous recovery, as shown in Figure R13g. To emulate spontaneous recovery using periodic weak reset pulses, we first applied consecutive 100 set pulses for habituation. Afterward, we withheld the set pulses for 100 seconds. During the stimulus absence, periodic weak reset pulses (-0.7 V and 2 μ s) were applied to the device every 10 seconds. The partial recovery of the device conductance could be observed after the stimulus withholding period, resembling the spontaneous recovery in biological systems. These results demonstrate that the device can simulate spontaneous recovery through the use of periodic reset pulses, if such characteristics are required by the application system. We would like to emphasize that in robotic applications, where devices typically rely on battery power and have limited power sources, it is crucial to maintain stored information without degradation when powered off.

6) *Potentiation of habituation*

Potentiation of habituation refers to the phenomenon where the degree or rate of habituation increases with repeated series of stimuli, when periods of spontaneous recovery are alternated between stimulus series. To test the potentiation of habituation in the developed device, we applied two series of consecutive set pulses, with 100 seconds of stimulus withholding period between each habituation phase. During the withholding period, periodic weak reset pulses were applied every 10 seconds to induce pseudo spontaneous recovery. As shown in Figure R13h, faster habituation was observed after 100 seconds of pseudo spontaneous recovery. The results demonstrate that the device can emulate potentiation of habituation through the application of periodic reset pulses, simulating the effects of spontaneous recovery.

Figure R13. Synaptic behaviors associated with habituation. **a.** Dishabituation. The 100 sets of positive voltage pulses (0.75 V and 5 μ s) were applied to the device, resulting in the habituation. After the habituation, consecutive 10 combined negative (-0.9 V and 1 μ s) and positive (0.75 V and 5 μ s) voltage pulses corresponding to pain and tactile stimuli, respectively, were applied to the device. The presentation of the combined pulses increased the device conductance back to LRS, indicating the dishabituation. **b.** Habituation of dishabituation. The 100 sets of positive voltage pulses (0.75 V and 5 μ s) were applied to the device for habituation. After the first habituation, consecutive 100 dishabituation combined voltage pulses (-0.9 V, 1 μ s and 0.75 V, 5 μ s) were applied to the habituated device. The device conductance initially increased by the dishabituation, but it soon decreased as the dishabituation pulses continued, indicating the habituation against the dishabituation stimuli. **c** and **d.** Stimulus frequency dependency. To investigate the effect of stimuli frequency, two positive voltage pulses (0.75 V, 3 μ s) with different pulse intervals (from 1 μ s to 10 ms) were repetitively applied to the device (**c**). The device exhibited the stimulus frequency dependency with faster habituation and lower final output current for shorter pulse intervals or higher frequency (**d**). **e** and **f.** Stimulus intensity dependency. Consecutive 100 positive set voltage pulses with different amplitudes of 0.75, 1.00, 1.05, 1.10 V and width of 5 μ s were applied to the device (**e**). The device exhibited pronounced habituation for the 0.75 V case. However, the higher pulse amplitude resulted in higher output currents, due to thicker filament formation (**f**). The results demonstrate the stimulus intensity dependency of the third-order memristor. **g.** Spontaneous recovery. Spontaneous recovery requires time-decaying memory or volatile memory. To emulate spontaneous recovery in the non-volatile third-order memristor, periodic weak reset pulses (-0.7 V, 2 μ s) were utilized. After 100 sets of positive voltage pulses (0.75 V, 5 μ s) for habituation, positive voltage pulses were withheld for 100 seconds. During this withholding period, the periodic weak reset

pulses were applied to the device every 10 seconds. Partial recovery of the habituated state was observed after the 100 seconds of withholding period, demonstrating the pseudo-spontaneous recovery. **h.** Potentiation of habituation. Two stages of habituation events were presented (0.75 V and 5 μ s), with the 100 seconds of stimulus withholding period. During the withholding period, the periodic weak reset pulses were applied for pseudo-spontaneous recovery. It was observed that the second habituation was faster than the first habituation, demonstrating the potentiation of habituation. Here, the device output current was measured by the read pulse (0.3 V and 100 μ s).

To clarify the habituation characteristics of the developed device with various synaptic behaviors such as dishabituation, we have appended Figure R13 in the revised Supplementary Information as Supplementary Figure S19. The corresponding explanations have been appended in Supplementary Information as Supplementary Note S1. Furthermore, we have compared synaptic behaviors of previously reported memristor devices in Table R2 to highlight the superior habituation characteristics of the proposed device. The appended explanations in the revised manuscript are as follows:

Page 11, line 207: “It is noteworthy that there are some discrepancies between the habituation of the third-order memristor and that of biological synapses. While biological synapses exhibit spontaneous recovery due to short-term memory, the third-order memristor features non-volatile memory. The absence of short-term memory may limit the use of the third-order memristor for accurately emulating synaptic behaviors in biological SNS for neuroscience studies. However, reliable robotic applications often require learned information to persist for a long time or even when powered off, making non-volatile memory advantageous. Despite the absence of short-term memory, the third-order memristor demonstrated various synaptic behaviors associated with habituation, as shown in Supplementary Figs. S19, S20, and Supplementary Note S1. Furthermore, certain biological synaptic behaviors reliant on short-term memory, such as spontaneous recovery and potentiation of habituation, can be mimicked by periodically applying weak reset pulses, if necessary (see Supplementary Figs. S19g, S19h, and S20). Detailed comparisons of the third-order memristor with several devices emulating habituation, as presented in Supplementary Table S1, highlight its favorable electrical characteristics for reliable robotic applications, as well as ability to mimic various synaptic behaviors.”

Ref.	1	2	3	4	5	6	7	This work
Device	TiN/ Li _x SiO _y /Pt	Au/LiTaO ₃ /Pt	Pt/LLTO /Pt	Pt/SNO/Pt	Pt/NiO/Pd	W/HfO ₂ / Ti/TiN	Al/HfO ₂ / IGZO/Au	TiN/HfO ₂ /TiO ₂ / Ti/TiN
CMOS Compatibility	No	No	No	No	No	Yes	No	Yes
Retention	10,000 s	>1,000 s	<200 s	12 h	20 s	N/A	N/A	10 years at 440.42 K
Cycle-to-cycle uniformity	N/A	N/A	N/A	N/A	2%	N/A	N/A	LRS: 3.9% HRS: 21.1%
Device-to- device uniformity	N/A	N/A	N/A	N/A	N/A	N/A	N/A	LRS: 7% HRS: 27%
Switching speed	100 ns	1 μs	0.4 s	200 s	0.5 s	1 ms	50 ms	250 ns
Habituation	Yes	Yes	Yes	Yes	Yes	Yes	Yes	Yes
Dishabituation	Yes	No	Yes	No	No	Yes	No	Yes
Habituation of dishabituation	Yes	No	Yes	No	No	No	No	Yes
Stimulus intensity dependence	Yes	Yes	Yes	No	Yes	Yes	Yes	Yes
Stimulus frequency dependence	Yes	Yes	Yes	No	Yes	Yes	Yes	Yes
Potentiation of habituation	Yes	No	No	No	No	No	No	Yes
Spontaneous recovery	Yes	No	Yes	Yes	Yes	No	Yes	Yes
Application	Robot navigation (Simul.)	N/A	N/A	Spiking neural network (Simul.)	Homeostatic learning (Simul.)	N/A	N/A	Neuro-inspired robot arm control (Exp.)

Table R2. Comparison with various memristors emulating habituation characteristics. To demonstrate the advantages of the third-order memristor for emulating habituation characteristics, several key characteristics are compared. For utilizing a memristor-based artificial sensory nervous system (MASNS) in robotic applications, non-volatile memory, good uniformity, fast switching speed, as well as ability to emulate various habituation characteristics are required. The comparison reveals the third-order switching complexity enables non-volatile habituation state and various habituation characteristics, while CMOS-compatible HfO₂-based resistive switching offers good uniformity and fast switching speeds.

Comment #2:

The learning should be dependent on the stimulus strength, I do not see this discussed, please clarify.

Response: We sincerely appreciate the reviewer for raising important concerns. We agree that demonstrating stimulus strength dependency is essential for assessing device characteristics as an artificial synapse in MASNS. Therefore, we measured the device conductance changes in response to repeated pulses with varying frequency and intensity to assess stimulus frequency dependency and stimulus intensity dependency, respectively (see Figure R14). Additionally, we analyzed the effect of pain stimulus strength on dishabituation by varying negative voltage pulse amplitudes (see Figure R15).

As shown in Figure R14a, device conductance update curves were obtained by applying 50 sets of two voltage pulses (0.75 V, 3 μ s) with different pulse intervals (1 μ s to 10 ms). The final current output for each case is compared in Figure R14b. The device exhibited faster habituation and lower final output current for shorter pulse interval (higher frequency) cases. This frequency dependency originates from the thermal effect of the device, where a shorter pulse interval generates larger Joule heat with less heat dissipation. The different final output currents and different habituation behaviors in response to the pulse frequency demonstrate that the learning in the device depends on the stimulus frequency.

Similarly, the effect of stimulus intensity was analyzed by applying consecutive 100 voltage pulses with different pulse amplitudes (0.75, 1.00, 1.05, and 1.10 V, 5 μ s). As shown in Figure R14c and R14d, the device showed pronounced habituation with a lower final output current at a pulse amplitude of 0.75 V. However, with higher pulse amplitudes, the device showed higher final output currents compared to the 0.75 V case. The increased device conductance for higher pulse amplitudes of 1.00, 1.05, and 1.10 V can be attributed to the formation of a thicker filament due to the stronger applied electric field. These results demonstrate that the device exhibits stimulus intensity dependency.

Figure R14. Stimulus frequency and intensity dependency of the third-order memristor. **a.** Stimulus frequency dependency. Consecutive 50 sets of two voltage pulses were applied to the device while varying the pulse interval. More pronounced habituation was observed for shorter pulse interval, or higher frequency. **b.** Output current of the device according to the stimulus interval after applying 100 pulses. **c.** Stimulus intensity dependency. Consecutive 100 voltage pulses were applied to the device while varying the pulse amplitude. While lower pulse amplitude induced lower current and faster habituation, higher pulse amplitudes resulted in higher output current due to thick filament formation. **d.** Output current of the device according to the stimulus intensity after applying 100 pulses.

The effects of the pain stimulus strength, or the negative voltage pulse amplitude, were investigated, as shown in Figure R15. After the first habituation process with repeated positive voltage pulses, a dishabitation pulse composed of negative (-0.5 to -0.8 V, 1 μ s) and positive voltage pulses (0.75 V, 5 μ s) was applied to the device, followed by the second habituation process. In the dishabitation pulse, the negative voltage pulse amplitude represents the strength of the pain stimulus, while the positive voltage pulse amplitude represents the strength of the tactile stimulus. As shown in Figures R15a and R15b, the device exhibited negligible dishabitation when the pain pulse amplitudes were low (-0.5 and -0.6 V). However, the dishabitation became more pronounced when the negative voltage pulse amplitude reached -0.7 V. The strongest negative voltage pulse with -0.8 V resulted in a clear dishabitation with a large conductance change. The results demonstrate that the dishabitation in the developed device is affected by the stimulus strength, similar to the habituation cases.

Figure R15. Effects of the negative voltage pulse amplitude on dishabitation. Dishabitation stimulus with combined voltage pulses was applied to the device after habituation. **a.** Dishabitation with -0.5 V and 1 μ s of negative voltage pulse. **b.** Dishabitation with -0.6 V and 1 μ s of negative voltage pulse. **c.** Dishabitation with -0.7 V and 1 μ s of negative voltage pulse. **d.** Dishabitation with -0.8 V and 1 μ s of negative voltage pulse. As the larger negative voltage pulse amplitude, which represents the strength of the pain stimulus, is applied, the more pronounced dishabitation is observed. In this experiment, positive voltage pulses of 0.75 V and 5 μ s and read voltage pulses of 0.3 V and 100 μ s were utilized.

We have revised the Supplementary Information to demonstrate the effects of the stimulus strength on the habituation and dishabitation. Figure R14 is appended as Supplementary Figure S19c-f to provide clarifications of the stimulus frequency and intensity dependency. Figure R15 is appended as Supplementary Figure S25 to demonstrate the effect of the stimulus strength on dishabitation. The corresponding explanations are appended in the revised manuscript as follows:

Page 11, line 207: “It is noteworthy that there are some discrepancies between the habituation of the third-order memristor and that of biological synapses. While biological synapses exhibit spontaneous recovery due to short-term memory, the third-order memristor features non-

volatile memory. The absence of short-term memory may limit the use of the third-order memristor for accurately emulating synaptic behaviors in biological SNS for neuroscience studies. However, reliable robotic applications often require learned information to persist for a long time or even when powered off, making non-volatile memory advantageous. Despite the absence of short-term memory, the third-order memristor demonstrated various synaptic behaviors associated with habituation, as shown in Supplementary Figs. S19, S20, and Supplementary Note S1. Furthermore, certain biological synaptic behaviors reliant on short-term memory, such as spontaneous recovery and potentiation of habituation, can be mimicked by periodically applying weak reset pulses, if necessary (see Supplementary Figs. S19g, S19h, and S20). Detailed comparisons of the third-order memristor with several devices emulating habituation, as presented in Supplementary Table S1, highlight its favorable electrical characteristics for reliable robotic applications, as well as ability to mimic various synaptic behaviors.”

Page 16, line 289: “The synaptic characteristics, such as the number of stimuli required for sensitization or degree of dishabituation against a threatening event, can be finely adjusted by modifying the pulse condition for each stimulus, as shown in Supplementary Fig. S24 and S25, respectively. The conductance change of the device according to the type of applied stimuli enables habituation to the safe stimuli while sensitization to the threatening stimuli.”

Comment #3:

Sensitization memory timescale is dependent on the stimulus strength. This is not shown anywhere.

Response: We sincerely thank the reviewer for this constructive comment. As the reviewer mentioned, a biological synapse showed varying sensitization memory timescales depending on stimulus strength. In biological systems, a stronger sensitization stimulus leads to longer sensitization memory. However, the developed device possesses non-volatile memory without time-dependent decaying, which makes it difficult to directly exhibit sensitization memory timescales depending on stimulus strength.

However, as demonstrated in our previous response in comment #1, it is possible to implement a form of spontaneous recovery by using periodic weak reset pulses, if the decaying property is required for a specific application. To investigate this, we applied sensitization pulses to the device and measured its conductance over a period of 1,000 seconds, while applying pseudo spontaneous recovery through weak reset pulses (-0.7 V, 2 μ s) every 10 seconds (Figure R16). As shown in Figure R16a, the device exhibited more rapid sensitization and higher conductance with stronger stimulus pulses. Moreover, the stronger pulse amplitudes resulted in a different sensitization memory timescale due to the slower pseudo spontaneous recovery process. The final current after 1,000 seconds of pseudo spontaneous recovery demonstrated that the device conductance remained higher for cases with stronger pulse amplitude (see Figure R16b). These results demonstrate that the different sensitization memory timescale, which depends on stimulus strength, can be emulated through the pseudo spontaneous recovery process if needed.

Figure R16. Sensitization memory timescale according to the stimulus strength. a. Sensitization memory timescale with pseudo spontaneous recovery. After applying two positive voltage pulses with various amplitudes (0.7, 0.725, 0.75, 0.775, and 0.8 V, 5 μ s), stimulus withholding was presented with pseudo spontaneous recovery (-0.7 V and 2 μ s per 10 seconds). While periodic weak reset pulse gradually reduced conductance, it was observed that the higher stimulus intensity resulted in higher final conductance after 1,000 seconds. **b.** Output current of the device according to the stimulus intensity after 1,000 seconds of stimulus withholding.

However, we would like to emphasize once again that our work focused on improving the efficiency of robotic systems based on memristor-based artificial sensory nervous system (MASNS) with habituation, rather than attempting to replicate exact synaptic behaviors in neuroscience that may not be suitable for real-world applications.

To clarify the non-volatile memory characteristics of the device, we have appended corresponding explanations in the revised manuscript. In addition, we have appended Figure R16 in the revised Supplementary Information as Supplementary Figure S20 to demonstrate that the spontaneous recovery property of a biological synapse could be emulated in such non-volatile memory devices by applying periodic reset pulses. The detailed revisions are presented as follows:

Page 11, line 207: “It is noteworthy that there are some discrepancies between the habituation of the third-order memristor and that of biological synapses. While biological synapses exhibit spontaneous recovery due to short-term memory, the third-order memristor features non-volatile memory. The absence of short-term memory may limit the use of the third-order memristor for accurately emulating synaptic behaviors in biological SNS for neuroscience studies. However, reliable robotic applications often require learned information to persist for a long time or even when powered off, making non-volatile memory advantageous. Despite the absence of short-term memory, the third-order memristor demonstrated various synaptic behaviors associated with habituation, as shown in Supplementary Figs. S19, S20, and Supplementary Note S1.”

Page 11, line 214: “Furthermore, certain biological synaptic behaviors reliant on short-term memory, such as spontaneous recovery and potentiation of habituation, can be mimicked by periodically applying weak reset pulses, if necessary (see Supplementary Figs. S19g, S19h, and S20). Detailed comparisons of the third-order memristor with several devices emulating habituation, as presented in Supplementary Table S1, highlight its favorable electrical characteristics for reliable robotic applications, as well as ability to mimic various synaptic behaviors.”

Comment #4:

Typical habituation involves forgetting as the stimulus is presented at different intervals. In this case, if there is no change in memory, then I am not sure this is habituation, but rather just a drop in the strength of the e-field across the layer that is changing the property (e.g conductance).

Response: We thank the reviewer for this constructive comment. As the reviewer mentioned, biological synapses exhibit forgetting when stimuli are presented at different intervals, due to their short-term memory characteristics. Therefore, if the goal of a study is to develop a strictly biologically plausible artificial synaptic device for neuroscience applications, it would be necessary to replicate all synaptic behaviors observed in biological systems. However, we would like to highlight that the primary focus of our work is on developing a novel device that exhibits habituation-like behaviors for efficient robotic systems and demonstrating the effectiveness of robotic systems with MASNS, rather than developing biologically plausible artificial synapses in the context of neuroscience study.

In addition, we would like to clarify that the observed drop in conductance under identical pulse conditions can be interpreted as habituation in the developed MASNS. Habituation refers to the reduction of response to repeated identical stimuli, and this mechanism is advantageous for living organisms by reducing response to insignificant stimuli. In the gill-withdrawal reflex in *Aplysia*, synaptic depression of sensory-motor synapses induces habituation¹². In our study, the memristor functions as an artificial synapse, where its conductance represents the synaptic strength between the pre-synaptic sensory neuron and a post-synaptic motor neuron. In this context, the drop of the device conductance, initially increased under identical stimulus conditions, mimics the reduction in synaptic strength observed in biological sensory nervous systems during habituation to repeated stimuli (see Figure R17a).

It is noteworthy that conventional memristor devices exhibit monotonic conductance update curves, where the device conductance initially increases and saturates when identical pulses are repeated (see Figure R17b). Therefore, conventional memristors are not suitable for emulating habituation characteristics due to this monotonic conductance update property. The conductance update curves of the third-order memristor and a conventional low-order memristor, shown in Figure R17c and R17d, respectively, highlight the distinct switching characteristics observed in response to repeated stimuli.

Figure R17. Comparisons of the third-order memristor to conventional low-order memristors. **a.** Illustration of the conductance update curve of the third-order memristor. **b.** Illustration of the conductance update curve of conventional low-order memristors, showing monotonic conductance update curve with conductance saturation. **c.** Example of the conductance update curve from the third-order memristor, demonstrating non-monotonic conductance change. **d.** Example of the conductance update curve from the low-order memristor, showing monotonic conductance update and conductance saturation.

We have appended Figure R17 in the revised Supplementary Information as Supplementary Figure S1 to emphasize different conductance update characteristics of the developed third-order memristor compared to conventional low-order memristors.

Page 7, line 129: “These low-order memristors have been hardly used for implementing the habituation of the biological SNS, because the device conductance usually monotonically increases with set pulses (red line of Fig.1d and Supplementary Fig. S1).”

Comment #5:

It is entirely not clear what the purpose of the robot arm is in this study, since the motion of this arm seems simply dependent on the strength of the stimulus. Is there a new capability that arises? Isn't the response obvious in terms of less signal strength in, less response amplitude out?

Response: We are sincerely grateful to the reviewer for this constructive comment. The robot arm system was developed to demonstrate how system efficiency could be enhanced by utilizing the MASNS with the third-order memristor. In the robot arm system, the strength of the applied stimuli or the pulse amplitude remained constant. Despite the identical input conditions, the MASNS adapted to repeated stimuli and optimized the robot arm's response without requiring bulky peripheral circuitry, mimicking the behavior of biological sensory nervous systems in living organisms.

Notably, the rotate motor received different signals based on the output current from the memristor, analogous to a motor neuron receiving different excitatory post-synaptic potential (EPSP) from synapses in biological systems. The MASNS improved the efficiency of the robotic system by diminishing responses to repetitive and insignificant stimuli, while maintaining sensitivity to significant or threatening stimuli, as demonstrated in our robot arm experiments.

The unique non-monotonic conductance update of the third-order memristor offers a new capability for robotic applications. As shown in Figure R18a, in conventional robotic systems without MASNS, all the received repeated stimuli from sensors are transmitted to a processor in conventional robotic systems without MASNS. This results in the processor handling large amounts of data, leading to increased energy consumption and throughput limitations. In contrast, in a robotic system with MASNS proposed in this work, the third-order memristor discriminates between repeated, insignificant stimuli through habituation, transmitting a reduced signal to the processor (see Figure R18b). This approach enables the processor to handle only important stimuli, resulting in low energy consumption and reduced latency.

The robot arm system experimentally validated the effectiveness of incorporating MASNS with third-order memristors in robotic systems. We would like to highlight that few studies have experimentally demonstrated habituation behaviors in robotic systems with MASNS, due to challenges such as low device-to-device and cycle-to-cycle uniformities, slow switching speeds, or low yield.

Figure R18. Illustrations of the effectiveness of the MASNS for robotic systems. a. Conventional robotic system without MASNS. In the conventional robotic system, the processor receives all the signals from sensors, resulting in large energy consumption and throughput limitations. **b.** Neuro-inspired robotic system with MASNS. In the robotic system with MASNS, the MASNS with third-order memristor filters out insignificant stimuli. The processor in this system processes only important stimuli. Therefore, the energy efficiency and latency of the system can be improved.

To clarify the goal of the robot arm system demonstration, we have appended Figure R18 in the revised Supplementary Information as Supplementary Figure 26. The revised parts of the manuscript are as follows:

Page 17, line 297: “**Implementing MASNS in robotic systems could improve energy efficiency while reducing processor burden by effectively filtering out insignificant external signals, as shown in Supplementary Fig. S26.** To investigate the effectiveness of the third-order memristor for an intelligent MASNS, we built a MASNS-implemented robot arm system (see Supplementary Fig. S27 and Methods). **The robot arm generates movements in response to external stimuli that produce a memristor output current exceeding a specific threshold, analogous to the response of a motor system to sensory stimulation that generates excitatory post-synaptic potentiation exceeding a threshold potential in biological SNS (see Supplementary Fig. S28).** The reaction of the robot arm against external stimuli (touching and electric shock) was tested with both the low-order and the third-order memristors (see Fig. 5).”

Comment #6:

So this is simply saturation or self-limiting due to reduced voltage being dropped in the dielectric, rather than habituation. In which case, it is not clear if much of the discussion in the Introduction etc is relevant to the paper.

Response: We sincerely appreciate the reviewer for this valuable comment. In this work, habituation refers to the depression of response or sensory-motor synaptic strength in response to repeated stimuli. However, as the reviewer pointed out in previous comments, other synaptic behaviors beyond synaptic depression, such as dishabituation, should be demonstrated to substantiate the claim that the proposed device exhibits habituation.

To address this, we have demonstrated that the developed device can emulate various synaptic behaviors relevant to habituation, including habituation, dishabituation, habituation of dishabituation, stimulus frequency/intensity dependency, spontaneous recovery, and potentiation of habituation, as shown in Figure R19. The results confirm that the device exhibits habituation characteristics, rather than a simple synaptic depression.

Figure R19. Synaptic behaviors associated with habituation. **a.** Dishabituation. The 100 sets of positive voltage pulses (0.75 V and 5 μ s) were applied to the device, resulting in the habituation. After the habituation, consecutive 10 combined negative (-0.9 V and 1 μ s) and positive (0.75 V and 5 μ s) voltage pulses corresponding to pain and tactile stimuli, respectively, were applied to the device. The presentation of the combined pulses increased the device conductance back to LRS, indicating the dishabituation. **b.** Habituation of dishabituation. The 100 sets of positive voltage pulses (0.75 V and 5 μ s) were applied to the device for habituation. After the first habituation, consecutive 100 dishabituation combined voltage pulses (-0.9 V, 1 μ s and 0.75 V, 5 μ s) were applied to the habituated device. The device conductance initially increased by the dishabituation, but it soon decreased as the dishabituation pulses continued, indicating the habituation against the dishabituation stimuli. **c** and **d.** Stimulus frequency dependency. To investigate the effect of stimuli frequency, two positive voltage pulses (0.75 V, 3 μ s) with different pulse intervals (from 1 μ s to 10 ms) were repetitively applied to the device (**c**). The device exhibited the stimulus frequency dependency with faster habituation and lower final output current for shorter pulse intervals or higher frequency (**d**). **e** and **f.** Stimulus intensity dependency. Consecutive 100 positive set voltage pulses with different amplitudes of 0.75, 1.00, 1.05, 1.10 V and width of 5 μ s were applied to the device (**e**). The device exhibited pronounced habituation for the 0.75 V case. However, the higher pulse amplitude resulted in higher output currents, due to thicker filament formation (**f**). The results demonstrate the stimulus intensity dependency of the third-order memristor. **g.** Spontaneous recovery. Spontaneous recovery requires time-decaying memory or volatile memory. To emulate spontaneous recovery in the non-volatile third-order memristor, periodic weak reset pulses (-0.7 V, 2 μ s) were utilized. After 100 sets of positive voltage pulses (0.75 V, 5 μ s) for habituation, positive voltage pulses were withheld for 100 seconds. During this withholding period, the periodic weak reset pulses were applied to the device every 10 seconds. Partial recovery of the habituated state was observed after the

100 seconds of withholding period, demonstrating the pseudo-spontaneous recovery. **h.** Potentiation of habituation. Two stages of habituation events were presented (0.75 V and 5 μ s), with the 100 seconds of stimulus withholding period. During the withholding period, the periodic weak reset pulses were applied for pseudo-spontaneous recovery. It was observed that the second habituation was faster than the first habituation, demonstrating the potentiation of habituation. Here, the device output current was measured by the read pulse (0.3 V and 100 μ s).

In addition, comparisons between the third-order memristor and conventional low-order memristors demonstrate that conventional memristors exhibit simple conductance saturation or self-limiting behavior, which is not observed in the third-order memristor (see Figure R20). As shown in Figure R20a, the typical conductance update curve of a conventional low-order memristor displays a monotonic increase in conductance followed by saturation or self-limiting. This monotonic behavior originates from the low-order switching dynamics, where the growth or rupture of a conductive filament driven by the applied electric field dominates the conductance change. The simple monotonic conductance update and saturation of conventional low-order memristors are not suitable for MASNS due to the lack of habituation functionalities.

In contrast, the third-order memristor exhibits a unique non-monotonic conductance update, where the conductance initially increases but then decreases as stimuli are repeated (see Figure R20b). This non-monotonic curve originates from complex switching dynamics, where both filament growth and TiO_x layer conductance contribute to the overall device conductance. While some memristor devices have demonstrated habituation, they often rely on volatile memory characteristics, limiting their use in real-world robotic applications. On the contrary, the developed third-order memristor possesses stable retention characteristics with non-volatile memory, good uniformity, high yield, fast switching speeds, and large-scale CMOS-compatibility. The favorable device performances make the third-order memristor highly suitable for reliable robotic applications.

Figure R20. Comparisons of the third-order memristor to conventional low-order memristors. a. Conductance update curve from the low-order memristor, showing monotonic conductance update and conductance saturation. **b.** Conductance update curve from the third-order memristor, demonstrating its unique non-monotonic conductance update curve.

To clarify that the device exhibits habituation characteristics with various synaptic behaviors, we have appended Figure R19 in the revised Supplementary Information as Supplementary Figure S19 with corresponding explanations. We have also appended Figure R20 in the revised Supplementary Information as Supplementary Figure S1 to emphasize the different electrical

characteristics of the third-order memristor compared to conventional memristors. The revised parts of the manuscript are as follows:

Page 11, line 207: “It is noteworthy that there are some discrepancies between the habituation of the third-order memristor and that of biological synapses. While biological synapses exhibit spontaneous recovery due to short-term memory, the third-order memristor features non-volatile memory. The absence of short-term memory may limit the use of the third-order memristor for accurately emulating synaptic behaviors in biological SNS for neuroscience studies. However, reliable robotic applications often require learned information to persist for a long time or even when powered off, making non-volatile memory advantageous. Despite the absence of short-term memory, the third-order memristor demonstrated various synaptic behaviors associated with habituation, as shown in Supplementary Figs. S19, S20, and Supplementary Note S1. Furthermore, certain biological synaptic behaviors reliant on short-term memory, such as spontaneous recovery and potentiation of habituation, can be mimicked by periodically applying weak reset pulses, if necessary (see Supplementary Figs. S19g, S19h, and S20). Detailed comparisons of the third-order memristor with several devices emulating habituation, as presented in Supplementary Table S1, highlight its favorable electrical characteristics for reliable robotic applications, as well as ability to mimic various synaptic behaviors.”

Page 7, line 129: “These low-order memristors have been hardly used for implementing the habituation of the biological SNS, because the device conductance usually monotonically increases with set pulses (red line of Fig.1d and Supplementary Fig. S1). ”

Comment #7:

Please discuss prior literature on habituation / sensitization experiments using memory devices and replication of early neuroscience studies including use of identical material as in the current manuscript:

Zuo, F, et al. "Habituation based synaptic plasticity and organismic learning in a quantum perovskite." *Nature communications* 8.1 (2017): 240.

Jiang, R, et al. "Habituation/Fatigue behavior of a synapse memristor based on IGZO–HfO₂ thin film." *Scientific Reports* 7.1 (2017): 9354.

Mondal, S, et al. "All-electric nonassociative learning in nickel oxide." *Advanced Intelligent Systems* 4.10 (2022): 2200069.

Yang, X, et al. "Nonassociative learning implementation by a single memristor-based multi-terminal synaptic device." *Nanoscale* 8.45 (2016): 18897-18904.

Response: We are truly grateful to the reviewer for this insightful suggestion. As the reviewer commented, it is essential to compare the developed third-order memristor to previous studies that have reported devices for habituation and sensitization characteristics. To address the reviewer's comments, we have compared the proposed device to previously reported devices reporting habituation. The results of these comparisons are shown in Table R3.

As shown in Table R3, several devices have been proposed to emulate habituation characteristics for MASNS, neural networks with homeostasis, or neuroscience studies. To utilize MASNS for robotic applications, a memristor should meet various criteria, including CMOS-compatibility, non-volatile memory with long retention, good uniformity, fast switching speed, and the ability to exhibit habituation characteristics. However, various studies have focused more on mimicking biological synaptic properties rather than experimentally demonstrating robotic applications based on MASNS.

For example, memristor devices based on Li_xSiO_y or LiTaO₃ demonstrated biologically plausible habituation characteristics with habituation, dishabituation, stimulus intensity dependence, stimulus frequency dependence, or spontaneous recovery^{1,2}. However, these devices lack analysis on CMOS-compatibility and uniformity, which hinders their utilization in reliable robotic applications.

Other previous studies reported memristor devices based on LiLaTiO (LLTO), SmNiO₃ (SNO), or NiO for emulating habituation characteristics³⁻⁵. However, these devices exhibited volatile memory with short retention, meaning that trained information fades after some time. In addition, their slow switching speeds (>0.4 s) degrade the response speed of the developed MASNS.

Furthermore, there have been memristor devices with habituation characteristics using the HfO_x layer, which is the same material used in our device^{6,7}. The use of the CMOS-compatible HfO_x layer enhances integration compatibility with current CMOS technology, enabling the integration of memristors on Si substrates with circuitry. Although one of the materials used, HfO_x, is identical, these devices have demonstrated monotonic conductance updates of low-order memristors. Due to the monotonic conductance update of low-order memristors, the demonstration of certain important habituation characteristics, including dishabituation or

habituation of dishabituation, was missed. We would like to emphasize that the material in different devices with different stacks, structures, or fabrication processes, can lead to significantly different electrical or chemical characteristics, resulting in entirely different device characteristics.

To assess the advantages of the proposed third-order memristor compared to these previous studies, we measured several electrical performance metrics that are required for real-world applications, including retention, switching speeds, cycle-to-cycle uniformity, and device-to-device uniformity (Figure R21). The results demonstrated that the proposed third-order memristor possesses non-volatile retention (10 years retention at 440 K), good cycle-to-cycle uniformity (σ/μ of 0.039, 0.211 for LRS, HRS, respectively), good device-to-device uniformity (σ/μ of 0.07, 0.27 for LRS, HRS, respectively), and fast switching speeds (250 ns), as well as CMOS-compatibility and habituation characteristics. These favorable characteristics allow the developed memristor to experimentally demonstrate robotic applications based on MASNS.

To discuss and compare various devices emulating habituation, we have appended Table R3 and Figure R21 as Supplementary Table S1 and Supplementary Figure S5, respectively, in the revised Supplementary Information. In addition, we have appended corresponding explanations by modifying the manuscript as follows:

Page 9, line 149: “The fabricated HfO₂ memristor exhibited favorable operation characteristics with **stable retention, fast switching speed**, low switching variations, and good endurance (Supplementary Fig. S5 and S6). **Furthermore, the high yield within 8-inch large-scale CMOS-compatible fabrication (Supplementary Figs. S7 and S8) and high thermal stability in elevated temperatures (Supplementary Fig. S9) enable integration of the device with sensor arrays and make the device suitable for reliable robotic applications in various environments.**”

Page 11, line 217: “**Detailed comparisons of the third-order memristor with several devices emulating habituation, as presented in Supplementary Table S1, highlight its favorable electrical characteristics for reliable robotic applications, as well as ability to mimic various synaptic behaviors.**”

Ref.	1	2	3	4	5	6	7	This work
Device	TiN/ Li _x SiO _y /Pt	Au/LiTaO ₃ /Pt	Pt/LLTO /Pt	Pt/SNO/Pt	Pt/NiO/Pd	W/HfO ₂ / Ti/TiN	Al/HfO ₂ / IGZO/Au	TiN/HfO ₂ /TiO ₂ / Ti/TiN
CMOS Compatibility	No	No	No	No	No	Yes	No	Yes
Retention	10,000 s	>1,000 s	<200 s	12 h	20 s	N/A	N/A	10 years at 440.42 K
Cycle-to-cycle uniformity	N/A	N/A	N/A	N/A	2%	N/A	N/A	LRS: 3.9% HRS: 21.1%
Device-to- device uniformity	N/A	N/A	N/A	N/A	N/A	N/A	N/A	LRS: 7% HRS: 27%
Switching speed	100 ns	1 μs	0.4 s	200 s	0.5 s	1 ms	50 ms	250 ns
Habituation	Yes	Yes	Yes	Yes	Yes	Yes	Yes	Yes
Dishabituation	Yes	No	Yes	No	No	Yes	No	Yes
Habituation of dishabituation	Yes	No	Yes	No	No	No	No	Yes
Stimulus intensity dependence	Yes	Yes	Yes	No	Yes	Yes	Yes	Yes
Stimulus frequency dependence	Yes	Yes	Yes	No	Yes	Yes	Yes	Yes
Potentiation of habituation	Yes	No	No	No	No	No	No	Yes
Spontaneous recovery	Yes	No	Yes	Yes	Yes	No	Yes	Yes
Application	Robot navigation (Simul.)	N/A	N/A	Spiking neural network (Simul.)	Homeostatic learning (Simul.)	N/A	N/A	Neuro-inspired robot arm control (Exp.)

Table R3. Comparison with various memristors emulating habituation characteristics. To demonstrate the advantages of the third-order memristor for emulating habituation characteristics, several key characteristics are compared. For utilizing a memristor-based artificial sensory nervous system (MASNS) in robotic applications, non-volatile memory, good uniformity, fast switching speed, as well as ability to emulate various habituation characteristics are required. The comparison reveals the third-order switching complexity enables non-volatile habituation state and various habituation characteristics, while CMOS-compatible HfO₂-based resistive switching offers good uniformity and fast switching speeds.

Figure R21. Favorable electrical characteristics of the third-order memristor for large-scale real-world robotic applications. **a.** Retention characteristics of the device showing stable and non-volatile memory with 10 years retention at approximately 440 K. The retention was extracted by monitoring fail time at elevated temperatures (543, 553, 565, and 573 K) and extrapolating the fitted results. The extracted activation energy was 0.97 eV, close to the previously reported activation energy of oxygen vacancies in HfO_2 . **b.** Fast switching speeds of the device. The device exhibited abrupt resistive switching and habituation for pulse widths longer than 500 ns. For pulse widths as short as 250 ns, the device displayed resistive switching without habituation. The set pulse amplitude was 0.85 V, and the device was measured by read pulses (0.3 V and 100 μs). **c** and **d.** Cycle-to-cycle variation of the device. Consecutive 100 I - V sweeps were applied to the device (**c**). The device exhibited stable and uniform bipolar resistive switching, with a variation coefficient (σ/μ) of 0.039 and 0.211 for LRS and HRS, respectively (**d**). **e** and **f.** Device-to-device variation of the device. The I - V curves from randomly selected 10 devices were compared (**e**). The randomly selected devices exhibited similar electrical characteristics without severe device-to-device variation, with a variation coefficient of 0.07 and 0.27 for LRS and HRS, respectively (**f**).

Comment #8:

The authors present a Methods section on modeling of the device, but it is not clear how this is relevant to the paper. Is the model used instead of the device to move the robot? Since there exist numerous papers already on both TiO_x and HfO₂, is this model providing any new information or validates some previous hypothesis etc?

Response: We are sincerely grateful to the reviewer for this constructive comment. We agree that there was a lack of explanations regarding the relationship between the developed device model and the study. As the reviewer pointed out, numerous studies provide models for conventional low-order memristors, such as HfO₂ or TiO_x based memristors. However, it should be noted that these models represent only conventional low-order memristors with monotonic conductance update curves, and do not account for the third-order memristor developed in this study. Therefore, we have developed a new device model consisting of two anti-serially connected resistive switching elements, the conductive filament and the TiO_x layer, to confirm that the resistive switching of these two elements induces the non-monotonic conductance update characteristics. The modeling results demonstrate that the anti-serial connection of two switching elements in a single device leads to the non-monotonic conductance update behavior, which is a distinctive feature not commonly observed in conventional memristor devices.

In addition, we would like to clarify that the robot experiments were conducted using the fabricated device, not based on simulations of the model. The physical third-order memristor device was directly employed as an artificial synapse in the robot arm with MASNS, modulating the response of the robot arm to external stimuli similar to biological synapses.

We have revised the manuscript to clarify that the model was developed to confirm the non-monotonic conductance update induced by the anti-serial connection of two switching variables. Furthermore, we have also revised the Methods section to emphasize that the robot experiments were performed using the physical device, rather than a simulation based on the device model. The detailed revisions are provided as follows:

Page 13, line 228: “It matches well with the experimental data of the third-order memristor. **The identical non-monotonic conductance update curve from the model, which is hardly acquired in conventional low-order memristors, demonstrates that the habituation characteristics originate from the anti-serially connected resistive switching filament and TiO_x layer.**”

Page 21, line 397: “**The robot arm system with MASNS was built and experimentally tested to investigate the effectiveness of the third-order memristor-based MASNS in robotic applications.**”

Comment #9:

The authors claim the memristor response is equivalent to the sea slug's behavior, but this is clearly wrong, since there is no timescale involved in the device due to the non-volatile nature. Hence, it is entirely not clear if the authors have mis-interpreted the results of effective low field dropping across the O migration layer being mistaken for non-associative learning.

Response: We sincerely thank the reviewer for pointing out this important comment. The comparison to the sea slug's behavior was made to emphasize that some synaptic behaviors of the biological sensory nervous system, such as synaptic depression against repeated stimuli or synaptic potentiation against pain stimuli, can be emulated in the third-order memristor and utilized for improving the efficiency of robotic applications. We agree that claiming the developed MASNS behavior is equivalent to the sea slug's behavior could mislead the manuscript because there are no decaying or volatile memory characteristics of biological synapses in the developed memristor.

To avoid misleading and clarify that the developed MASNS has no time-decaying memory of the biological sensory nervous system in sea slugs, we have revised the manuscript as follows:

Page 16, line 292: **“The conductance change of the device according to the type of applied stimuli enables habituation to the safe stimuli while sensitization to the threatening stimuli.”**
(Previous sentence: “Notably, the measured memristor conductance trends for the three applied stimulus scenarios resemble the motor neuron activity of a sea slug's biological SNS, demonstrating that the developed MASNS exhibits similar synaptic characteristics to the biological SNS in terms of habituation and sensitization.”)

In addition, we would like to highlight that the conductance change in the third-order memristor can be regarded as a form of non-associative learning, such as habituation and sensitization, even in the absence of time-decaying memory observed in biological systems. The memristor in the MASNS acts as an artificial synapse between a pre-synaptic sensory neuron and a post-synaptic motor neuron, modulating the response of the system to applied stimuli. The oxygen migration in the device, induced by the applied stimuli, alters the device conductance, which in turn reflects changes in synaptic strength. When identical stimuli are repeatedly applied to a sensor, the memristor receives corresponding identical voltage pulses as pre-synaptic spikes. Initially, the voltage pulses increase the device conductance, resulting in sensitization of the MASNS. However, with continuous identical stimuli, the conductance gradually decreases, leading to habituation with reduced synaptic strength and reduced response.

Although time-decaying memory is not present in this study, the MASNS exhibits several key characteristics of habituation, including dishabituation, habituation of dishabituation, stimulus frequency dependency, and stimulus intensity dependency (see Figure R22). Furthermore, we have demonstrated that spontaneous recovery can be implemented using periodic weak reset pulses if the time-decaying memory is necessary for a specific application (see Figure R22g and R22h). The primary focus of our work is to develop MASNS with the CMOS-compatible and large-scale third-order memristor, with the aim of enhancing the efficiency of robotic applications.

Figure R22. Synaptic behaviors associated with habituation. **a.** Dishabituation. The 100 sets of positive voltage pulses (0.75 V and 5 μ s) were applied to the device, resulting in the habituation. After the habituation, consecutive 10 combined negative (-0.9 V and 1 μ s) and positive (0.75 V and 5 μ s) voltage pulses corresponding to pain and tactile stimuli, respectively, were applied to the device. The presentation of the combined pulses increased the device conductance back to LRS, indicating the dishabituation. **b.** Habituation of dishabituation. The 100 sets of positive voltage pulses (0.75 V and 5 μ s) were applied to the device for habituation. After the first habituation, consecutive 100 dishabituation combined voltage pulses (-0.9 V, 1 μ s and 0.75 V, 5 μ s) were applied to the habituated device. The device conductance initially increased by the dishabituation, but it soon decreased as the dishabituation pulses continued, indicating the habituation against the dishabituation stimuli. **c** and **d.** Stimulus frequency dependency. To investigate the effect of stimuli frequency, two positive voltage pulses (0.75 V, 3 μ s) with different pulse intervals (from 1 μ s to 10 ms) were repetitively applied to the device (**c**). The device exhibited the stimulus frequency dependency with faster habituation and lower final output current for shorter pulse intervals or higher frequency (**d**). **e** and **f.** Stimulus intensity dependency. Consecutive 100 positive set voltage pulses with different amplitudes of 0.75, 1.00, 1.05, 1.10 V and width of 5 μ s were applied to the device (**e**). The device exhibited pronounced habituation for the 0.75 V case. However, the higher pulse amplitude resulted in higher output currents, due to thicker filament formation (**f**). The results demonstrate the stimulus intensity dependency of the third-order memristor. **g.** Spontaneous recovery. Spontaneous recovery requires time-decaying memory or volatile memory. To emulate spontaneous recovery in the non-volatile third-order memristor, periodic weak reset pulses (-0.7 V, 2 μ s) were utilized. After 100 sets of positive voltage pulses (0.75 V, 5 μ s) for habituation, positive voltage pulses were withheld for 100 seconds. During this withholding period, the periodic weak reset

pulses were applied to the device every 10 seconds. Partial recovery of the habituated state was observed after the 100 seconds of withholding period, demonstrating the pseudo-spontaneous recovery. **h.** Potentiation of habituation. Two stages of habituation events were presented (0.75 V and 5 μ s), with the 100 seconds of stimulus withholding period. During the withholding period, the periodic weak reset pulses were applied for pseudo-spontaneous recovery. It was observed that the second habituation was faster than the first habituation, demonstrating the potentiation of habituation. Here, the device output current was measured by the read pulse (0.3 V and 100 μ s).

To clarify that the device possesses non-volatile memory characteristics, while the biological synapse has short-term memory with decaying property, we have revised the manuscript as follows:

Page 11, line 207: “It is noteworthy that there are some discrepancies between the habituation of the third-order memristor and that of biological synapses. While biological synapses exhibit spontaneous recovery due to short-term memory, the third-order memristor features non-volatile memory. The absence of short-term memory may limit the use of the third-order memristor for accurately emulating synaptic behaviors in biological SNS for neuroscience studies. However, reliable robotic applications often require learned information to persist for a long time or even when powered off, making non-volatile memory advantageous. Despite the absence of short-term memory, the third-order memristor demonstrated various synaptic behaviors associated with habituation, as shown in Supplementary Figs. S19, S20, and Supplementary Note S1. Furthermore, certain biological synaptic behaviors reliant on short-term memory, such as spontaneous recovery and potentiation of habituation, can be mimicked by periodically applying weak reset pulses, if necessary (see Supplementary Figs. S19g, S19h, and S20). Detailed comparisons of the third-order memristor with several devices emulating habituation, as presented in Supplementary Table S1, highlight its favorable electrical characteristics for reliable robotic applications, as well as ability to mimic various synaptic behaviors.”

Comment #10:

Overall, given the numerous literature on habituation learning (including with interval training) in various flavors of memristor devices including the exact material reported in this study, namely HfO₂, it is far from obvious what is the novel result or scientific insight in this manuscript.

Response: We sincerely appreciate the reviewer for raising this important comment. As the reviewer mentioned, several devices have been proposed to emulate habituation, and it is essential to compare the developed third-order memristor with previous studies to highlight the advancements in our work. To address the reviewer’s comment, we have provided Table R4 comparing the proposed third-order memristor with several previously reported devices emulating habituation.

As shown in Table R4, several devices have demonstrated habituation, or synaptic depression in response to repeated stimuli. However, many of these devices have primarily focused on realistically mimicking biological synaptic characteristics, often relying on volatile memory and exhibiting slow switching speeds. These limitations hinder the practical application of such

devices in real-world applications. Furthermore, the lack of key device performance metrics, such as uniformity, yield, and large-scale CMOS-compatible fabrication, further degrades the suitability of these devices for reliable applications. Consequently, the experimental demonstration of the MASNS with habituation has been rarely reported.

Ref.	1	2	3	4	5	6	7	This work
Device	TiN/ Li _x SiO _y /Pt	Au/LiTaO ₃ /Pt	Pt/LLTO /Pt	Pt/SNO/Pt	Pt/NiO/Pd	W/HfO _x / Ti/TiN	Al/HfO _x / IGZO/Au	TiN/HfO ₂ /TiO _x / Ti/TiN
CMOS Compatibility	No	No	No	No	No	Yes	No	Yes
Retention	10,000 s	>1,000 s	<200 s	12 h	20 s	N/A	N/A	10 years at 440.42 K
Cycle-to-cycle uniformity	N/A	N/A	N/A	N/A	2%	N/A	N/A	LRS: 3.9% HRS: 21.1%
Device-to- device uniformity	N/A	N/A	N/A	N/A	N/A	N/A	N/A	LRS: 7% HRS: 27%
Switching speed	100 ns	1 μs	0.4 s	200 s	0.5 s	1 ms	50 ms	250 ns
Habituation	Yes	Yes	Yes	Yes	Yes	Yes	Yes	Yes
Dishabituation	Yes	No	Yes	No	No	Yes	No	Yes
Habituation of dishabituation	Yes	No	Yes	No	No	No	No	Yes
Stimulus intensity dependence	Yes	Yes	Yes	No	Yes	Yes	Yes	Yes
Stimulus frequency dependence	Yes	Yes	Yes	No	Yes	Yes	Yes	Yes
Potentialion of habituation	Yes	No	No	No	No	No	No	Yes
Spontaneous recovery	Yes	No	Yes	Yes	Yes	No	Yes	Yes
Application	Robot navigation (Simul.)	N/A	N/A	Spiking neural network (Simul.)	Homeostatic learning (Simul.)	N/A	N/A	Neuro-inspired robot arm control (Exp.)

Table R4. Comparison with various memristors emulating habituation characteristics. To demonstrate advantages of the third-order memristor for emulating habituation characteristics, several key characteristics are compared. For utilizing a memristor-based artificial sensory nervous system (MASNS) in robotic applications, non-volatile memory, good uniformity, fast switching speed, as well as ability to emulate various habituation characteristics are required. The comparison reveals the third-order switching complexity enables non-volatile habituation state and various habituation characteristics, while CMOS-compatible HfO₂-based resistive switching offers good uniformity and fast switching speeds.

In addition, while the HfO₂ is widely used as a CMOS-compatible material for various devices, including memristors, transistors, and sensors, the device characteristics can vary significantly due to factors such as fabrication process, device stacks, or device structure. For instance, some previous studies have reported memristor devices incorporating the HfO₂ layer^{6,7}. However, these devices exhibited typical monotonic conductance updates of conventional memristors, differing from the non-monotonic conductance update observed in the developed third-order memristor. This difference in device behavior can be attributed to the inclusion of an additional resistive switching TiO_x layer in the third-order memristor, which arises from distinct fabrication conditions and device stacks.

To clarify the advancements of the developed device for MASNS and robotic applications, we have appended Table R4 as Supplementary Table S1 in the revised Supplementary Information. In addition, we have revised the manuscript to emphasize the advancements of the developed device compared to devices in previous studies as follows:

Page 9, line 149: “The fabricated HfO₂ memristor exhibited favorable operation characteristics with **stable retention, fast switching speed**, low switching variations, and good endurance (Supplementary Fig. S5 and S6). **Furthermore, the high yield within 8-inch large-scale CMOS-compatible fabrication (Supplementary Figs. S7 and S8) and high thermal stability in elevated temperatures (Supplementary Fig. S9) enable integration of the device with sensor arrays and make the device suitable for reliable robotic applications in various environments.**”

Page 11, line 217: “**Detailed comparisons of the third-order memristor with several devices emulating habituation, as presented in Supplementary Table S1, highlight its favorable electrical characteristics for reliable robotic applications, as well as ability to mimic various synaptic behaviors.**”

Comment #11:

Further, given the electrical data does not contain temporal information, it is questionable whether the results should be considered in the context of ‘habituation’ that is borrowed from neuroscience or simply due to reduced current or voltage being applied to the second layer.

Response: We are truly grateful to the reviewer for this insightful comment. We agree that the proposed device and the MASNS exhibit certain differences from biological synapses and biological sensory nervous systems. The primary goal of our work is to develop the third-order memristor and the artificial sensory nervous system to emulate specific functions that enhance the efficiency of robotic applications, rather than to replicate all aspects of biological synapses for neuroscience studies. From this perspective, we refer to the process as habituation because it represents the adjustment of synaptic strength in response to repeated stimuli, optimizing reactions for efficiency. While this process may lack the time-decaying properties observed in biological synapses, it still captures the essential characteristics necessary for robotic applications.

To address the reviewer’s concerns, we have clarified that the behavior of the device has some differences from biological synapses in certain aspects, such as time-decaying memory property, in the revised manuscript. Additionally, we have appended Supplementary Note S1 to emphasize that, while the device can emulate various synaptic behaviors associated with habituation, its non-volatile memory characteristics can be advantageous for reliable robotic applications, even though volatile memory is preferred for biological plausibility.

Page 11, line 207: “It is noteworthy that there are some discrepancies between the habituation of the third-order memristor and that of biological synapses. While biological synapses exhibit spontaneous recovery due to short-term memory, the third-order memristor features non-volatile memory. The absence of short-term memory may limit the use of the third-order memristor for accurately emulating synaptic behaviors in biological SNS for neuroscience studies. However, reliable robotic applications often require learned information to persist for a long time or even when powered off, making non-volatile memory advantageous. Despite the absence of short-term memory, the third-order memristor demonstrated various synaptic behaviors associated with habituation, as shown in Supplementary Figs. S19, S20, and Supplementary Note S1. Furthermore, certain biological synaptic behaviors reliant on short-term memory, such as spontaneous recovery and potentiation of habituation, can be mimicked by periodically applying weak reset pulses, if necessary (see Supplementary Figs. S19g, S19h, and S20). Detailed comparisons of the third-order memristor with several devices emulating habituation, as presented in Supplementary Table S1, highlight its favorable electrical characteristics for reliable robotic applications, as well as ability to mimic various synaptic behaviors.”

Page 28, line 501 (Supplementary Information):

“5) Spontaneous recovery

Spontaneous recovery refers to the recovery of a response after a period of stimulus absence after habituation. In several previous studies, spontaneous recovery has been realized in volatile

memory characteristics of devices, where trained information naturally fades due to the short retention of the memory. The non-volatile memory of the third-order memristor might be considered a discrepancy compared to biological synapses. While volatile memory can simulate spontaneous recovery, it is challenging to develop non-volatile or long-term memory of trained information in such volatile memory devices. In contrast, the non-volatile memory offers advantages in terms of reliability and energy-efficiency for robotic applications, as short-term memory devices would lose trained information over time or when powered off, making them unsuitable for such applications.

Furthermore, spontaneous recovery in non-volatile devices can be simulated by applying periodic weak reset pulses, which we termed pseudo spontaneous recovery, as shown in Supplementary Fig. S19g. To emulate the spontaneous recovery using periodic weak reset pulses, we first applied consecutive 100 set pulses for habituation. Afterward, set pulses were withheld for 100 seconds, and weak negative voltage reset pulses were applied to the device every 10 seconds during the stimulus absence. The partial recovery of the device conductance could be observed after the stimulus withholding period, resembling the spontaneous recovery in biological systems. These results demonstrate that the device can simulate spontaneous recovery through the use of periodic reset pulses, if such characteristics are required by the application system. However, it is noteworthy that, since various robotic applications use batteries and have limited power sources, it is crucial to maintain stored information without decaying when powered off, rather than accurately mimic biological synaptic behaviors.”

Reviewer #3

The authors experimentally demonstrate a third-order memristor-based artificial sensory nervous system for neuro-inspired robotics, which is an interesting work. However, the following questions should be addressed before the publication.

Response: We sincerely appreciate the reviewer for providing insightful comments on our work and for suggesting methods to improve the clarity of the study. As the reviewer suggested, we have conducted additional experiments to investigate the effects of environmental temperature and parasitic capacitance. We have also performed further material studies to better understand the switching mechanisms of the device. Our detailed responses to the reviewer's comments are provided below.

Comment #1:

The author attributes the effect of longer pulse time entirely to the change in temperature caused by Joule heat. Is there any evidence for this? I agree that higher temperatures would speed up the ion migration. But the repeated pulses with longer pulse time can also enhance the ion migration even if the temperature is fixed especially in the HRS state.

Response: We are truly grateful to the reviewer for this insightful comment. As the reviewer pointed out, there was a lack of evidence regarding the significant effect of temperature on the habituation characteristics. To address the reviewer's comment, we have analyzed the resistive switching characteristics at elevated temperatures.

As shown in Figure R23, the switching of the device was measured at various set voltage pulse amplitudes and temperatures of 300, 373, and 473 K. At room temperature (300 K), the device exhibited habituation when the pulse amplitude exceeded 0.8 V. At 373 K, a more pronounced habituation was observed, indicating that temperature significantly influences the habituation characteristics. Furthermore, at 473 K, habituation occurred at a lower pulse amplitude of 0.75 V, which is an amplitude that did not induce habituation at lower temperatures. These results demonstrate that the high temperature, which is generated by Joule heat effects during the device operations, significantly affects the switching in TiO_x layer and habituation characteristics.

To provide further evidence of the heat effect on the device switching, we have appended Figure R23 in the revised Supplementary Information as Supplementary Figure S18 with corresponding explanations as follows:

Page 11, line 203: “**In addition, measurements at elevated temperatures revealed more pronounced and fast habituation characteristics, demonstrating the effects of the high temperature on resistive switching in the TiO_x layer and habituation characteristics (see Supplementary Fig. S18). The results support the conclusion that the device temperature acts as a critical state variable in the developed third-order memristor.**”

Figure R23. Pulsed response of the device with consecutive 100 set pulses followed by 100 reset pulses while varying set voltages and temperatures. **a.** Pulsed response of the device at room temperature (300 K). Habituation characteristics were observed for the 0.8 V of set pulse amplitude. **b.** Pulsed response of the device at an elevated temperatures of 373 K, showing pronounced habituation. **c.** Pulsed response of the device at an elevated temperatures of 473 K, showing faster habituation at lower set pulse amplitudes (0.75 V), as well as pronounced habituation.

Comment #2:

It should be noted that the device at a high resistance state should act as a capacitor. When the device changes from HRS to LRS, there is possibly a capacitor discharge current in addition to conducting current, which may lead to "overshoot" current (like habituation behavior). This should be verified.

Response: We sincerely appreciate the reviewer for this constructive suggestion. As the reviewer commented, the effect of the intrinsic capacitance of the device should be verified. To clarify the effect of the intrinsic capacitance, we measured the device capacitance by applying AC small signal to the device before and after forming, as shown in Figure R24. For the device before the forming process, the device resistance is high enough to measure the intrinsic capacitance (see Figure R24a). Therefore, the intrinsic capacitance of approximately 60 fF was confirmed (see Figure R24b). On the other hand, the device after the forming process possesses very small electrical resistance compared to the device before the forming process, and therefore, the effect of the intrinsic capacitance is negligible (see Figure R24c). As shown in Figure R24d, the intrinsic capacitance of the device after forming could not be measured due to the small electrical resistance of the device, meaning that the intrinsic capacitance is negligible. The results show that the effect of the intrinsic device capacitance can be minimal due to the small electrical resistance after the forming process.

To further verify whether the effect of the intrinsic capacitance and overshoot current affect the device switching, we have measured device switching while varying the pulse interval between set voltage pulse and read pulse. Since the current overshoot from intrinsic capacitance and RC delay depends on the pulse interval, the effect of the intrinsic capacitance can be confirmed by comparing the results of various pulse interval cases. As shown in Figure R25, the habituation characteristics have been measured while varying the pulse interval between the set voltage pulse and the read pulse from 100 μ s to 10 ms. It can be observed that there are no differences for all the pulse interval cases, demonstrating the effect of the intrinsic capacitance is negligible and the current overshoot is minimal in the developed device. The results prove that the habituation characteristics with non-monotonic conductance update is a unique device switching characteristics, rather than a non-ideal current overshoot phenomenon due to the intrinsic capacitance.

Figure R24. Intrinsic capacitance of the device before and after forming. **a.** Schematic of the device with intrinsic capacitance and high electrical resistance before forming. Due to the high electrical resistance, the effect of the intrinsic capacitance could be dominant before forming. **b.** Results of capacitance measurement before forming. **c.** Schematic of the device with intrinsic capacitance and low electrical resistance after forming. Since the electrical resistance after the forming process is small, the effect of the intrinsic capacitance becomes minimal. **d.** Results of capacitance measurement after forming, showing measurement failure due to the low electrical resistance of the device.

Figure R25. Pulsed response of the device with extended pulse intervals. To minimize the possible effect of the current overshoot and RC delay, the device was measured by increasing pulse intervals between set pulse and read pulse. Similar output current characteristics for all the pulse interval cases demonstrate that the current overshoot and RC delay hardly affect the device characteristics.

We have appended Figure R24 and Figure R25 in the revised Supplementary Information as Supplementary Figures S14 and S15, respectively, to clarify that the intrinsic capacitance of the device and current overshoot hardly affect the switching characteristics. The corresponding explanations have been appended in the revised manuscript as follows:

Page 10, line 182: “Furthermore, we investigated whether the habituation characteristics originate from the device’s resistive switching or the transient and parasitic effects, such as intrinsic capacitance or current overshoot. By analyzing the intrinsic capacitance of the device and varying the pulse interval to mitigate current overshoot, as shown in Supplementary Figs. S14 and S15, it was confirmed that the habituation characteristics arise from the resistive switching in the device, rather than from transient or parasitic effects.”

Comment #3:

During the sensitization process, the number of pulses (experienced in 1st stage of Fig.S8(a)) is minimal, which is inconvenient to the application. Can the sensitization process be well controlled by more pulse numbers? It is better to demonstrate this by adjusting pulse off time or amplitude.

Response: We truly thank the reviewer for providing this important comment. As the reviewer mentioned, the number of pulses that induce maximum sensitization should be controllable to be utilized in several applications. As suggested by the reviewer, adjusting the pulse width or amplitude can be utilized to modify the number of pulses for maximum sensitization. Reducing the pulse width or amplitude results in the increased pulse number required for maximum sensitization, as shown in Figure R26. However, reducing the pulse width or amplitude also slows down the habituation, resulting in a minimal resistive change in the habituation stage.

Figure R26. Conductance update of the device with various set pulse conditions. a. The conductance update

curves of the third-order memristor for consecutive 100 set pulses (0.75 V and 2, 3, 4, and 5 μ s, respectively) followed by 100 reset pulses (-0.9 V and 1 μ s) are measured. **b.** The conductance update curves of the third-order memristor for consecutive 100 set pulses (0.7, 0.75, 0.8, and 0.85 V and 1 μ s, respectively) followed by 100 reset pulses (-0.9 V and 1 μ s) are measured. The habituation characteristic becomes evident when the set pulse width is longer (**a**) or when the set pulse amplitude is higher (**b**), demonstrating that enough set pulse amplitude and width are necessary to transfer the oxygen anions into the TiO_x layer. The device conductance is obtained by a read pulse (0.3 V and 50 μ s) in both (**a**) and (**b**) after applying each set and reset pulse.

To modulate the required pulse number for sensitization while enhancing the habituation characteristics, we found that utilizing a heating pulse could be an effective solution. A heating pulse is a low voltage pulse that generates Joule heating effect in the device without changing its conductance state (see Figure R27a). We applied the heating pulse followed by the programming voltage pulse (0.75 V, 1 μ s), where the programming voltage pulse condition was weak enough that it alone could not induce habituation. When the device is in the high resistance state (HRS), the generated Joule heat by the heating pulse is minimal, and therefore, the programming voltage pulse solely affects the resistive switching. This low Joule heat effect prevents abrupt sensitization, making a gradual sensitization process. On the other hand, when the device is in the low resistance state (LRS), the generated Joule heat by the heating pulse becomes significant. In this case, the generated Joule heat and the programming voltage pulse both affect the resistive switching. Therefore, the Joule heat enhances the device switching, enabling habituation characteristics despite the low programming voltage pulse amplitude. In this manner, introducing the heating pulse could be able to realize gradual sensitization with a larger number of pulses, while maintaining pronounced habituation characteristics.

As shown in Figure R27b-e, the conductance update curves of the device with various heating pulse amplitudes from 0.4 V to 0.6 V were measured while other conditions were fixed. When there is no heating pulse, as shown in Figure R27b, the device exhibited monotonic conductance update without habituation, due to the low and short programming pulse amplitude and width, respectively. However, when the heating pulse with 0.4 V of amplitude and 5 μ s of width was applied, the device showed habituation characteristics, even though the programming voltage pulse remained unchanged (see Figure R27c). As the heating pulse amplitude increases, more pronounced habituation could be observed, demonstrating the effect of the heating pulse on the device switching characteristics (see Figure R27d and R27e). Compared to results obtained with a higher programming voltage pulse (0.85 V, 1 μ s) but without the heating pulse, the utilization of the heating pulse and lower programming voltage pulse showed a gradual sensitization process while maintaining similar habituation characteristics (see Figure R27f). These findings demonstrate that the number of pulses required for sensitization can be effectively modulated through the heating pulse and the Joule heat effect, while retaining the habituation characteristics.

Figure R27. Using a heating pulse to adjust the number of set pulses for sensitization. **a.** Illustration of the heating pulse. A heating pulse is a low voltage pulse that generates Joule heat effect in the device, while not changing the device conductance. **b-e.** Pulsed responses of the device when a heating pulse of 0, 0.4, 0.5, and 0.6 V and 5 μ s, respectively, were used. The results without heating pulse exhibited gradual sensitization without habituation characteristics (**b**), while the application of heating pulse induced habituation characteristics (**c**, **d**, and **e**). The Joule heat effects from the heating pulse facilitate the oxidation of the TiO_x layer in the device, resulting in habituation characteristics. **f.** Comparison of the habituation characteristics with and without the heating pulse. When the high set voltage pulse (0.85 V and 1 μ s) was utilized without a heating pulse, abrupt sensitization was observed, where only three pulses were required for maximum sensitization. On the contrary, when the low set voltage pulse (0.75 V and 1 μ s) was used with a heating pulse (0.5 V and 5 μ s), more gradual sensitization was observed while showing similar habituation characteristics. The utilization of a heating pulse can be advantageous for adjusting the number of stimuli for sensitization without changing the habituation characteristics.

To clarify the adjustability of the number of pulses required for sensitization, we have appended Figure R27 in the revised Supplementary Information as Supplementary Figure S24. We have also revised the manuscript to clarify that the required number of pulses for sensitization could be adjusted, enhancing the suitability of the device for various applications.

Page 16, line 289: “The synaptic characteristics, such as the number of stimuli required for sensitization or degree of dishabituation against a threatening event, can be finely adjusted by modifying the pulse condition for each stimulus, as shown in Supplementary Fig. S24 and S25, respectively. The conductance change of the device according to the type of applied stimuli enables habituation to the safe stimuli while sensitization to the threatening stimuli.”

Comment #4:

What is the initial state in the TiO_x? As shown in Fig.3(c), the conductance change in the TiO_x layer is very small. Can the device work if no TiO_x layer in the device? The switching mechanism of the device is still unclear.

Response: We are truly grateful to the reviewer for providing this insightful comment. To address the reviewer’s comment regarding the initial state of the TiO_x layer and the detailed device switching mechanism, we have analyzed the stoichiometry of the TiO_x layer by using X-ray photoelectron spectroscopy (XPS), measured the switching characteristics of a device without TiO_x layer, and investigated the conduction mechanism of the device in each conductance state to disclose the exact switching mechanism. The detailed explanations are shown below:

1) Initial state of the TiO_x layer

We prepared two samples with and without the high temperature process (673 K, 4 minutes) equivalent to the back-end-of-line (BEOL) process conditions. XPS depth-profile tests were done on each sample, as shown in Figure R28. For the device without high temperature process, no evident TiO_x layer was observed between the Ti and HfO₂ layers (see Figure R28a). However, the device with the high temperature process exhibited the formation of the TiO_x layer, showing an evident shift in oxygen contents from HfO₂ to TiO_x (see Figure R28b). The results demonstrate that the high temperature environment in BEOL fabrication affects the formation of the TiO_x layer.

The binding energy of Ti 2p from the Ti-HfO₂ interface further demonstrates the initial state of the TiO_x layer. In the device without high temperature process, the Ti 2p spectra exhibited a peak binding energy of 454.57 eV, which is close to the binding energy of the binding energy of Ti metal (Ti⁰⁺, 454 eV)¹³, as shown in Figure R28c. On the contrary, the device with high temperature process exhibited a higher binding energy of 455.06 eV compared to that of the device without high temperature process¹³, as shown in Figure R28d. The extracted binding energy was close to that of the Ti²⁺ (455.1 eV), which demonstrates that the initial state of the TiO_x layer is close to TiO.

Figure R 28. XPS results of the device with and without high temperature process. a. Atomic percentage profiles of the device without high temperature process. The depth-profile results show that there is no evident TiO_x layer between the Ti and HfO₂ layers. **b.** Atomic percentage profiles of the device with high temperature process, demonstrating the formation of TiO_x layer between the Ti and HfO₂ layers with a clear shift of oxygen atoms to the Ti layer. **c.** The Ti 2p spectra of the Ti-HfO₂ interface in the device without high temperature process. The peak binding energy of 454.57 eV, which is close to the binding energy of Ti⁰⁺ (454 eV)¹³, demonstrates the metallic Ti is dominant in the Ti-HfO₂ interface without high temperature process. **d.** The Ti 2p spectra of the Ti-HfO₂ interface in the device with high temperature process, showing peak binding energy of 455.06 eV close to that of the Ti²⁺ (455.1 eV)¹³ and demonstrating the formation of TiO near the Ti-HfO₂ interface.

2) Device characteristics without the TiO_x layer

The switching characteristics of the device without the TiO_x layer were further analyzed to investigate the effects of the TiO_x layer on the habituation characteristics. As shown in Figure R29, the device without the high temperature process and having no TiO_x layer was tested by applying consecutive positive and negative voltage pulses. Since there is no TiO_x layer, the device exhibited a typical monotonic conductance update curve of the conventional low-order memristors. The results demonstrate that the TiO_x layer is essential for realizing the third-order switching and habituation characteristics.

Figure R29. Conductance update curve of the device without high temperature process. The device exhibited a monotonic conductance update curve similar to typical low-order memristors. The results demonstrate that the formation of the TiO_x layer via the high temperature BEOL process is essential for habituation characteristics.

3) Conduction mechanism analysis

We have compared the conduction mechanisms of the device in HRS, intermediate state between HRS and LRS, LRS, and habituation state, respectively. The I - V curves of the device in each state were measured. As shown in Figure R30a, the device in HRS, intermediate state, and LRS exhibited linear I - V characteristics, On the contrary, the habituation state exhibited non-linear I - V characteristics. The non-linear I - V curve of the habituation state demonstrates that the habituation state possesses different conduction mechanisms, despite the similar electrical resistivity to the intermediate state.

Typical oxide-based memristors rely on hopping conduction through oxygen defects. As shown in Figure R30b, HRS, intermediate state, and LRS exhibited hopping conduction with a linear relationship between current and voltage. On the other hand, the Schottky emission dominated the conduction in the habituation state, as shown in Figure R30c. The results demonstrate that the device in HRS, intermediate state, and LRS exhibits similar conduction mechanisms to a conventional low-order memristor, indicating that the conductive TiO_x layer contains a large number of oxygen vacancies (see Figure R30d, R30e, and R30f, respectively). However, as oxygen anions move to the TiO_x layer, the amount of oxygen vacancies in the TiO_x layer decreases and the insulating TiO_y layer is formed. The formation of an insulating TiO_y layer results in Schottky emission conduction due to a lower number of defects in the layer, leading to different conduction mechanisms compared to other states (see Figure R30g).

Figure R30. Conduction mechanisms of the device in each state. **a.** I - V characteristics of the device in HRS, intermediate state between HRS and LRS, LRS, and habituation state. **b.** Hopping conduction in HRS, intermediate state, and LRS state, similar to several oxide-based memristors. **c.** Schottky emission conduction in habituation state, demonstrating that the habituation state has different conduction mechanisms from other states. **d-g.** Switching mechanisms of the device based on the conduction mechanisms of each state. The device in HRS conducts electrons by hopping conduction, where the defect site density varies in the filament and the filament gap (**d**). As the filament growth, the conductivity of the device increases, showing intermediate state between HRS and LRS (**e**). The filament is fully grown in the LRS state, displaying the highest conductivity (**f**). As more set pulses are applied, oxygen anions from the filament to the TiO_x layer oxidize the TiO_x layer, resulting in the formation of an insulating and defectless TiO_y layer (**g**). Due to the lower defect density in the TiO_y layer, the device exhibits the Schottky emission, rather than hopping conduction.

The XPS results, electrical characteristics of the device without TiO_x layer, and the conduction mechanism analysis demonstrate that the TiO_x layer is an essential element for the habituation characteristics, and the reversible oxidation of the TiO_x layer is the critical mechanism in the device operation. To provide detailed information about the TiO_x layer and switching mechanisms, we have appended Figures R28, R29, and R30 in the revised Supplementary Information as Supplementary Figures S10, S11, and S21. We have also revised the manuscript as follows:

Page 9, line 155: “To understand the third-order resistive switching behavior based on the device structure, transmission electron microscopy (TEM) imaging, electron dispersion spectroscopy (EDS), X-ray photoelectron spectroscopy (XPS) results were obtained, as shown in Fig. 2b, 2c, and Supplementary Fig. S10. The TEM image in Fig. 2b shows metal-insulator-metal structure of the fabricated device, where the device consists of CMOS-compatible materials including the TiN, HfO₂, or Ti. It is noted that the device is fabricated in fully CMOS-compatible environments, which enables the monolithic integration of the device to various CMOS peripheral circuitries. The EDS line scan results, as shown in Fig. 2c, show the Ti/TiO_x/HfO₂ stack with an oxidized Ti-HfO₂ interface between the TiN electrodes. The initial stoichiometry of the TiO_x layer is close to TiO, as demonstrated by the XPS results (see Supplementary Fig. S10).”

Page 10, line 168: “The device without the TiO_x layer exhibited typical monotonic conductance update curve of conventional low-order memristors, demonstrating that the TiO_x layer is essential for habituation characteristics (see Supplementary Fig. S11).”

Page 13, line 232: “To further disclose the switching mechanism of the third-order memristor, the conduction mechanisms for each resistance state (HRS, intermediate state between HRS and LRS, LRS, and habituation state) were investigated, as shown in Supplementary Fig. S21. When the device was in HRS, intermediate state, or LRS, the linear *I-V* relationship of hopping conduction was confirmed (see Supplementary Fig. S21b). On the contrary, for the habituation state, the *I-V* curve exhibited a non-linear relationship, indicating that the Schottky emission dominates conduction in the habituation state (see Supplementary Fig. S21c). The different conduction mechanisms for each state can be explained by the initially conductive and defective TiO_x layer, where the oxygen anions from the filament reversibly oxidize the defective TiO_x layer into the resistive TiO_y layer with fewer defects, resulting in the Schottky emission (see Supplementary Figs. S21d-g). The results demonstrate that the movement of oxygen anions and the reversible oxidation of the TiO_x layer induce the habituation characteristic.”

References:

1. Wu, Z. *et al.* A habituation sensory nervous system with memristors. *Advanced Materials* **32**, 2004398 (2020).
2. Li, X. *et al.* Implementation of habituation on single ferroelectric memristor. *Appl. Phys. Lett.* **122**, 183505 (2023).
3. Shi, T., Wu, J. F., Liu, Y., Yang, R. & Guo, X. Behavioral plasticity emulated with lithium lanthanum titanate-based memristive devices: Habituation. *Adv. Electron. Mater.* **3**, 1700046 (2017).
4. Zuo, F. *et al.* Habituation based synaptic plasticity and organismic learning in a quantum perovskite. *Nat. Commun.* **8**, 240 (2017).
5. Mondal, S. *et al.* All-electric nonassociative learning in nickel oxide. *Adv. Intell. Syst.* **4**, 2200069 (2022).
6. Yang, X. *et al.* Nonassociative learning implementation by a single memristor-based multi-terminal synaptic device. *Nanoscale* **8**, 18897–18904 (2016).
7. Jiang, R., Ma, P., Han, Z. & Du, X. Habituation/Fatigue behavior of a synapse memristor based on IGZO-HfO₂ thin film. *Sci. Rep.* **7**, 9354 (2017).
8. Jiang, H., Li, C. & Xia, Q. Ta/HfO₂ memristors: From device physics to neural networks. *Jpn. J. Appl. Phys.* **61**, SM0802 (2022).
9. Suga, H. *et al.* Highly stable, extremely high-temperature, nonvolatile memory based on resistance switching in polycrystalline Pt nanogaps. *Sci. Rep.* **6**, 34961 (2016).
10. Pradhan, D. K. *et al.* Materials for high temperature digital electronics. *Nat. Rev. Mater.* **9**, 790–807 (2024).
11. Rankin, C. H. *et al.* Habituation revisited: An updated and revised description of the behavioral characteristics of habituation. *Neurobiol. Learn. Mem.* **92**, 135–138 (2009).
12. Byrne, J. H. Analysis of synaptic depression contributing to habituation of gill-withdrawal reflex in *Aplysia californica*. *J. Neurophysiol.* **48**, 431–438 (1982).
13. Peng, W. C. *et al.* Tunability of p- and n-channel TiO_x thin film transistors. *Sci. Rep.* **8**, 9255 (2018).

Response Letter to Reviewers' Comments

We truly appreciate the valuable time the reviewers have spent reviewing our revised manuscript and providing insightful comments and suggestions to help further improve the quality of our work. In this response letter, we have provided a point-by-point response to the reviewers' comments. We have appended two Supplementary Tables to compare power consumption of artificial synapses and to provide a detailed voltage and time conditions for synaptic operations in the revised Supplementary Information file. Furthermore, we have included discussions regarding potential challenges in large-scale integration and possible solutions to mitigate these issues. The detailed responses to each of the reviewers' comments are presented below.

Reviewer #1

I appreciate the authors' thorough responses to my previous queries. The additional experiments and analyses have significantly strengthened the manuscript.

Two areas still require attention:

Response: We are truly grateful to the reviewer for dedicating valuable time to carefully evaluate our manuscript and provide insightful comments. To address the reviewer's concerns, we have appended a table comparing power consumption of the third-order memristor to that of conventional circuit-based artificial synapses for sensory systems. This table highlights that the developed third-order memristor possesses superior energy efficiency compared to circuit-based systems, primarily due to the absence of bulky peripheral circuitry and complex processing steps required in conventional systems. Furthermore, we have included a discussion regarding the potential issues associated with large-scale integration, along with possible solutions to address these issues. The detailed responses regarding the reviewer's comments are presented below.

Comment #1:

While MASNS is claimed to improve energy efficiency, quantitative energy consumption data is lacking. Please provide measurements of the third-order memristor's power requirements across different operation modes (habituation, sensitization, standby) compared to conventional approaches.

Response: We deeply appreciate the reviewer for this insightful comment. As the reviewer pointed out, a quantitative comparison of the power consumption of the third-order memristor with conventional circuit-based artificial synapses for habituation is essential to evaluate the effectiveness of the proposed device. Accordingly, as shown in Table R1, we listed the power consumption values of conventional circuit-based systems for emulating non-associative learning (habituation and sensitization)¹ or associative learning², and compared them with the

power consumption of the third-order memristor for each synaptic operation (habituation, sensitization, and standby). Since experimental demonstrations of robotic systems with habituation implemented using conventional systems are scarce, direct comparison of the power consumption of complete robotic system is challenging. Therefore, we compared the power consumption of artificial synapses implementing habituation or associative learning through additional peripheral circuits to that of the third-order memristor, rather than analyzing the overall power consumption in the complete robotic system.

In conventional circuit-based systems for emulating non-associative or associative learnings, multiple circuit modules are required to assess the significance of the received stimuli and to apply specific voltage pulses for modulating the conductance of conventional memristors. The involvement of multiple circuit modules and processing steps results in increased power consumption and longer processing times. In contrast, the third-order memristor can emulate habituation characteristics without the need for peripheral circuitry, owing to its intrinsic non-monotonic conductance update characteristics. The absence of additional peripheral circuitry and multiple processing steps for emulating habituation leads to reduced power consumption and fast processing speeds.

	Ref.	Power (mW)	Pulse width for each processing (ms)
	1 (Habituation and sensitization)	12.06 (average)	~2
	2 (Associative learning)	7.78 (average)	2
This work	Habituation	3.94	0.001
	Sensitization (Dishabituation)	4.63	0.002
	Standby (Retaining synaptic state)	N/A	N/A

Table R1. Comparisons of power consumption and processing time of the third-order memristor and circuit-based systems. For circuit-based systems, multiple circuit modules are required for emulating synaptic functions such as non-associative or associative learnings. The need of multiple circuit modules and complex processing steps increase the power consumption and processing time. On the contrary, the third-order memristor exhibits habituation and sensitization characteristics based on its non-monotonic conductance update characteristics, without the need of multiple peripheral circuitries. The absence of complex peripheral circuitries and multiple processing steps results in the reduced power consumption and rapid processing time. Furthermore, due to the non-volatile memory characteristics of the third-order memristor, the synaptic state can be maintained during the standby or off state without the need of power-consuming refresh operations.

To address the reviewer’s comment, we have appended Table R1 in the revised Supplementary Information as Supplementary Table S3 with corresponding explanations as follows:

Page 12, Line 289: “**The realization of habituation and sensitization characteristics in the device without the need for bulky peripheral circuitry indicates the superior energy-efficiency of the**

third-order memristor-based MASNS for robotic applications. Supplementary Table S3 highlights the energy-efficiency of the third-order memristor-based MASNS compared to conventional circuit-based systems.”

Comment #2:

The manuscript demonstrates excellent individual device performance but insufficiently addresses large-scale integration challenges. Please discuss potential issues (crosstalk, thermal effects, etc.) when integrating multiple devices and propose mitigation strategies.

Response: We sincerely thank the reviewer for raising this important point. As the reviewer commented, there could be potential challenges associated with the large-scale integration of the third-order memristor.

For example, integrating the third-order memristor in a crossbar array structure could enable a high-density, large-scale array. However, due to the high electrical conductivity of the third-order memristor, a significant *IR*-drop could occur within the array, hindering the reliable operation of the devices³. This *IR*-drop issue would be further intensified under an extensive scaling on the crossbar array, owing to the increased wire resistance. To address this, designing metal wires with low resistance and limiting the size of a crossbar array could mitigate the *IR*-drop problem and enable reliable array operations. Additionally, reducing the overall device conductivity through material optimization could also be effective approach for suppressing the *IR*-drop issue.

Another potential issue is the sneak-path problem, where unintended current paths through unselected cells lead to interference and malfunctions within the array. This phenomenon also makes it difficult to accurately read the state of a targeted cell. To mitigate the sneak-path problem in a crossbar array, a selector device, such as ovonic threshold switch (OTS) or transistor, should be integrated in series with each memristor. Considering the analog characteristics of the third-order memristor, a transistor could serve as a suitable selector device, providing both precise cell selection and effective current regulation.

To address the reviewer’s comment, we have included a discussion on large-scale integration challenges and possible mitigation strategies in the revised manuscript as follows:

Page 15, Line 373: “It is noteworthy that potential issues may arise during large-scale array integration of the device, such as *IR*-drop due to the high device conductivity and sneak-path problems within the array. Material optimization to reduce the overall device conductivity, alongside the integration of appropriate selector devices, could mitigate these issues and facilitate the realization of large-scale integrated third-order memristor arrays.”

Reviewer #2

The authors have performed several additional experiments and provided clarifications. In response to one of the main questions concerning the analog between biology and their device, the authors state weak resetting pulses provide a pseudo-form of relaxation, since this is the closest they can come to in a non-volatile device. I assume this is a trade off between using a NVM device for memory versus synaptic learning and cannot be reconciled further with this device design. If the authors choose to, this may be worth discussing in the revised manuscript. I do not have further comments.

Response: We sincerely appreciate the time and effort the reviewer invested in evaluating and discussing our manuscript. The insightful comments provided by the reviewer have significantly contributed to improving the quality of the manuscript. To address the reviewer's suggestion, we have included a discussion regarding the trade-off between non-volatile memory characteristics and biological plausibility. Furthermore, we have provided voltage and time conditions of the device for various switching characteristics. The detailed point-by-point responses are provided below.

Comment #1:

I assume this is a trade off between using a NVM device for memory versus synaptic learning and cannot be reconciled further with this device design. If the authors choose to, this may be worth discussing in the revised manuscript. .

Response: We sincerely thank the reviewer for this important and thoughtful comment. As the reviewer pointed out, there exists a trade-off when using a non-volatile memory device for synaptic learning applications. Non-volatile memory devices offer superior synaptic weight stability, allowing the retention of synaptic states even when the power is turned off. However, many biological synaptic behaviors rely on time-decaying characteristics of volatile memory. Therefore, realistically mimicking biological synaptic learning becomes limited when employing non-volatile memory devices.

To address the reviewer's comment and to clarify this trade-off, we have revised the manuscript to include the following statement:

Page 9, Line 206: **“The use of non-volatile memory involves a trade-off between synaptic weight stability and biological plausibility. The absence of short-term memory may limit the use of the third-order memristor for accurately emulating synaptic behaviors for synaptic learnings.”**

Comment #2:

Furthermore, it is noted that various voltage / time conditions are reported that result in sample heating vs no heating and these are used in different training sequences as they report in the response file. It could be useful to prepare a simple table with the various voltage ranges that results in Joule heating or not and what values (or range of values) are used for the different training, reset conditions, etc. This would also be useful in case other researchers want to repeat some of this work on their own materials systems.

Response: We sincerely appreciate the reviewer for this insightful comment. As the reviewer commented, providing ranges of pulse voltages and widths for each operation mode would be helpful in evaluating the operation conditions of the device. To address the reviewer's comment regarding the pulse conditions, we have listed the pulse conditions and corresponding switching characteristics into a table, as shown in Table R2. Since the habituation process induced by set pulses was predominantly utilized in our work, while the reset process was only employed for device initialization, the effects of set pulse conditions are summarized in the table.

When the pulse amplitude was lower than 0.7 V, the third-order memristor exhibited no conductance switching when the pulse width was shorter than 1 μs , or showed a monotonic conductance update similar to low-order memristors, when the pulse width exceeded 1 μs . Under these conditions, habituation characteristics were barely observed due to the relatively low applied electric field.

For the pulse amplitudes between 0.7 and 0.8 V, the device exhibited varying switching characteristics depending on the pulse conditions. When the pulse width was shorter than 1 μs , the device typically exhibited a monotonic conductance update. However, when the pulse widths ranged from 1 to 3 μs , the device displayed either monotonic update or habituation characteristics, depending on the specific pulse amplitude, width, or interval. When the pulse width exceeded 3 μs , the device exhibited habituation characteristics.

At pulse amplitudes higher than 0.8 V, the device exhibited strong habituation characteristics under all pulse width conditions, attributed to the high electric field across the device. The provided table illustrates the relationship between the pulse conditions and switching characteristics, demonstrating that the device's switching behavior can be finely tuned by adjusting the pulse parameters.

Set pulse conditions	0.5-1 μ s	1-3 μ s	3-5 μ s
0.6-0.7 V	No switching	Monotonic update	Monotonic update
0.7-0.8 V	Monotonic update	Monotonic update or Habituation	Habituation
0.8-0.9 V	Habituation	Habituation	Habituation

Table R2. Effects of set pulse conditions on switching characteristics of the device. The device exhibited various switching characteristics, such as no switching, monotonic conductance update, or habituation characteristics, depending on the pulse conditions. Notably, when the pulse amplitude was 0.7-0.8 V and the width was 1-3 μ s, the device exhibited either monotonic update or habituation, depending on the specific pulse amplitude, width, or interval. The table demonstrates that the switching characteristics of the device can be finely tuned by adjusting the pulse conditions.

To address the reviewer’s comment regarding the clarification of pulse conditions and switching characteristics, we have appended Table R2 in the revised Supplementary Information as Supplementary Table S2 as follows:

Page 8, Line 179: “Conductance update curves of the device under various pulse conditions demonstrated that the third-order switching behavior requires a high amplitude or long width voltage pulse to relocate oxygen anions into the TiO_x layer (Supplementary Fig. S13 and Supplementary Table S2).”

References:

1. Hong, Q. *et al.* Memristive circuit implementation of biological nonassociative learning mechanism and its applications. *IEEE Transactions on Biomedical Circuit and Systems* **14**, 1036-1050 (2020).
2. Zhang, Y. *et al.* The framework and memristive circuit design for multisensory mutual associative memory networks. *IEEE Transactions on Cybernetics* **53**, 7844-7857 (2023).
3. Jeong, H. *et al.* Self-supervised video processing with self-calibration on an analogue computing platform based on a selector-less memristor array. *Nature Electronics* **8**, 168-178 (2025).